

# AutoWebWorld: Synthesizing Infinite Verifiable Web Environments via Finite State Machines

**Yifan Wu** [1 2] **Yiran Peng** [2] **Yiyu Chen** [1] **Jianhao Ruan** [1 2] **Zijie Zhuang** [1] **Cheng Yang** [2] **Jiayi Zhang** [1 2]
**Man Chen** [1] **Yenchi Tseng** [1] **Zhaoyang Yu** [2] **Liang Chen** [3] **Yuyao Zhai** [3] **Bang Liu** [4] **Chenglin Wu** [2] **Yuyu Luo** [1]

## Abstract

The performance of autonomous Web GUI agents heavily relies on the quality and quantity of training data. However, collecting high-quality interaction trajectories from real websites is expensive and difficult to verify. Specifically, the underlying state transitions of real websites are hidden from the agent: only rendered UI feedback (*e.g.*, screenshots) is observable, while the true internal state remains inaccessible. This forces existing pipelines to rely on external verifiers, such as human annotators or LLM judges, which are both inconsistent and costly. To address this, we propose **AutoWebWorld**, a framework for synthesizing controllable and verifiable web environments by modeling them as Finite State Machines (FSMs) and using coding agents to translate FSMs into *synthetic websites*. Unlike real websites, where state transitions are implicit, AutoWebWorld explicitly defines all states, actions, and transition rules. This enables programmatic verification: action validity is enforced by FSM preconditions and deterministic transition rules, and task success is certified by reaching a goal state in the FSM graph. Our approach automates the data generation pipeline, generating over 11,663 verified trajectories from 29 *synthetic websites* at only **$0.04 per trajectory**. Training on this synthetic data significantly boosts real-world performance. Our 7B Web GUI agent outperforms all baselines within 15 steps on WebVoyager. Furthermore, we observe a clear scaling law: as the synthetic data volume increases, performance on WebVoyager and Online-Mind2Web consistently improves.

[1]The Hong Kong University of Science and Technology(Guangzhou) [2]DeepWisdom [3]Peking University [4]Université de Montréal & Mila. Correspondence to: Bang Liu <bang.liu@umontreal.ca>, Chenglin Wu <alexanderwu@deepwisdom.ai>, Yuyu Luo <yuyuluo@hkust-gz.edu.cn>.

*Proceedings of the $43^{rd}$ International Conference on Machine Learning*, Seoul, South Korea. PMLR 306, 2026. Copyright 2026 by the author(s).

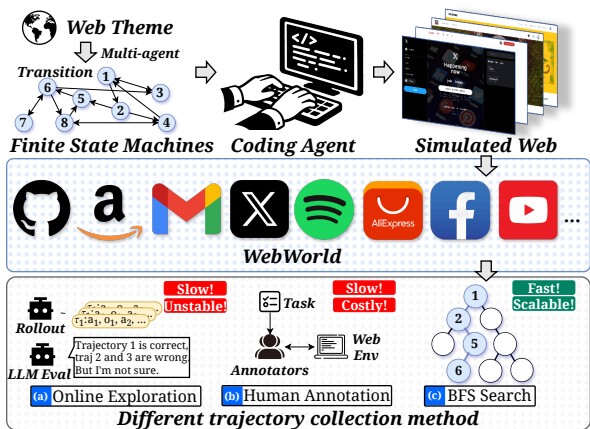

*Figure 1.* Comparison of GUI trajectory collection between existing methods and AutoWebWorld.

## 1. Introduction

Web GUI agents that autonomously navigate and interact with websites have emerged as a promising direction in AI (He et al., 2024; Awadallah et al., 2025; Zhu et al., 2025). While recent LLMs and MLLMs provide strong reasoning capabilities (Wang et al., 2024; Team et al., 2025; Wu et al., 2024), a primary bottleneck persists: the scarcity of high-quality, verifiable interaction trajectories for training (Pahuja et al., 2025; He et al., 2025; Lin et al., 2025).

A natural approach to obtaining training data is to collect trajectories from real-world websites, ensuring the training distribution matches deployment. As shown in Figure 1, existing methods for GUI trajectory synthesis primarily fall into two categories: *(a) online exploration* on live websites (He et al., 2024; Pahuja et al., 2025; Awadallah et al., 2025) and *(b) human demonstrations* (He et al., 2025; Deng et al., 2023). Despite their differences, all these methods are fundamentally *observation-based*: after each action, the agent only receives rendered UI feedback (e.g., a screenshot), while the true underlying state of the Web environment remains *hidden* (Zhou et al., 2023). For example, after clicking the button Add to cart, the Web UI may visually update, but whether the cart's internal state (e.g., item identity, quantity, applied discounts) has transitioned correctly

cannot be directly checked. Consequently, existing data collection methods rely on external verifiers, such as human annotators or LLM judges, to evaluate step-level correctness (Li et al., 2024; Lù et al., 2025; Gu et al., 2024; He et al., 2025). This forces verifiers to guess the internal state from surface observations, leading to a "*verifier bottleneck*" that is inconsistent (different verifiers often disagree), costly (human annotation and LLM judging are expensive).

To address the verifier bottleneck, we propose **AutoWebWorld**, a framework that unifies trajectory collection and verification in a controllable environment where the underlying state-transition logic is fully observable (Zhang et al., 2026a; 2025b). As shown in Figure 1, AutoWebWorld first leverages a multi-agent framework (Ruan et al., 2026) to generate an FSM based on a Web theme's name (*e.g.* Facebook or Gmail). This FSM explicitly specifies states, actions, and transition rules, pre-defining a corresponding GUI procedure for each action (Zhang et al., 2023). Data collection, therefore, shifts from *stochastic exploration* on real websites to *systematic Breadth-First Search* (BFS) over the known transition graph: we only expand actions whose preconditions are satisfied and follow deterministic transitions to obtain the shortest action sequence. For example, Add to cart is expanded only after BFS has traversed and fixed the concrete unit options of the product (*e.g.* size and color). We then synthesize a runnable front-end website from the same FSM, directly expand the BFS action sequence into executable GUI atomic operations, and replay them to collect screenshots while filtering trajectories that fail due to front-end implementation mismatches. This yields verified GUI trajectories without relying on human annotators or LLM judges.

To validate our approach, we implemented a library of **29** interactive web environments across diverse domains by running our AutoWebWorld. For data synthesis, we run the breadth-first search on the FSM state graphs to enumerate goal-reaching trajectories, and validate them by execution in the generated websites, resulting in *11,663 reproducibly verified trajectories* at the cost of only *$0.04 per trajectory*.

Training on this synthetic data yields significant real-world performance gains. An agent trained on merely 16K steps of our data achieves a state-of-the-art success rate of 27.42% on the challenging WebVoyager benchmark, outperforming models trained on datasets orders of magnitude larger. Furthermore, our results *provide strong evidence* that synthetic data can enable scaling for foundational models: as the amount of synthetic data increased, the performance of agents on real-world web benchmarks such as WebVoyager and Online-Mind2Web consistently improved, demonstrating the potential of synthetic data to drive real-world performance gains. Our contributions are threefold:

1. We propose **AutoWebWorld**, a state-driven paradigm

for synthesizing verifiable, scalable, and controllable web environments through an FSM-based foundation.

2. We release **29** diverse web environments and over 11,663 verified trajectories as both training data and a reproducible benchmark, and our pipeline can further scale up the environment and data volume at low cost.

3. We demonstrate best performance on WebVoyager within 15 steps and reveal a clear scaling law: increasing synthetic data consistently improves agent performance.

Our code is open-sourced at `https://evanwu1125.github.io/AWW_homepage/`.

## 2. Related Work

### 2.1. Data Scaling for GUI Agent

To obtain high-quality GUI interaction trajectories for training, prior work primarily collects data from real environments (Deng et al., 2023; Lù et al., 2024). Existing approaches can be grouped into three categories. First, task-driven online exploration places an agent on real websites, executes tasks from natural language instructions, and records interaction trajectories (Pahuja et al., 2025; Murty et al., 2024; Trabucco et al., 2025; Awadallah et al., 2025). Second, human demonstrations collect expert executions of tasks in real environments as supervision (He et al., 2025). Third, video-based methods extract interaction signals from human screen recordings and convert them into structured action sequences (Lu et al., 2025; Zhang et al., 2025a). Despite their different data sources, these pipelines usually cannot scale cleanly because correctness is underspecified in real environments and must be determined by external verifiers (Pan et al., 2024a). In practice, this requires post hoc judging and filtering of actions and trajectories by humans or learned evaluators, which introduces cost and inconsistency (Xu et al., 2024). This verifier bottleneck, therefore, becomes the key limit for scaling high-quality trajectories.

### 2.2. Synthetic GUI Environment

Scaling trajectory collection on real websites is difficult largely because correctness must be determined by external verifiers (Zhou et al., 2023; Pan et al., 2024a). A natural alternative is to embed verification into the environment so that success criteria are defined and checked internally (Shi et al., 2017; Yao et al., 2022). In practice, most synthetic or semi-synthetic web environments have been developed primarily as evaluation benchmarks (Chezelles et al., 2024; Liu et al., 2018) with more stable verification and reproducible execution pipelines compared to online web benchmarks (Xue et al., 2025; He et al., 2024; Pan et al., 2024b). These benchmarks span templated task suites and packaged

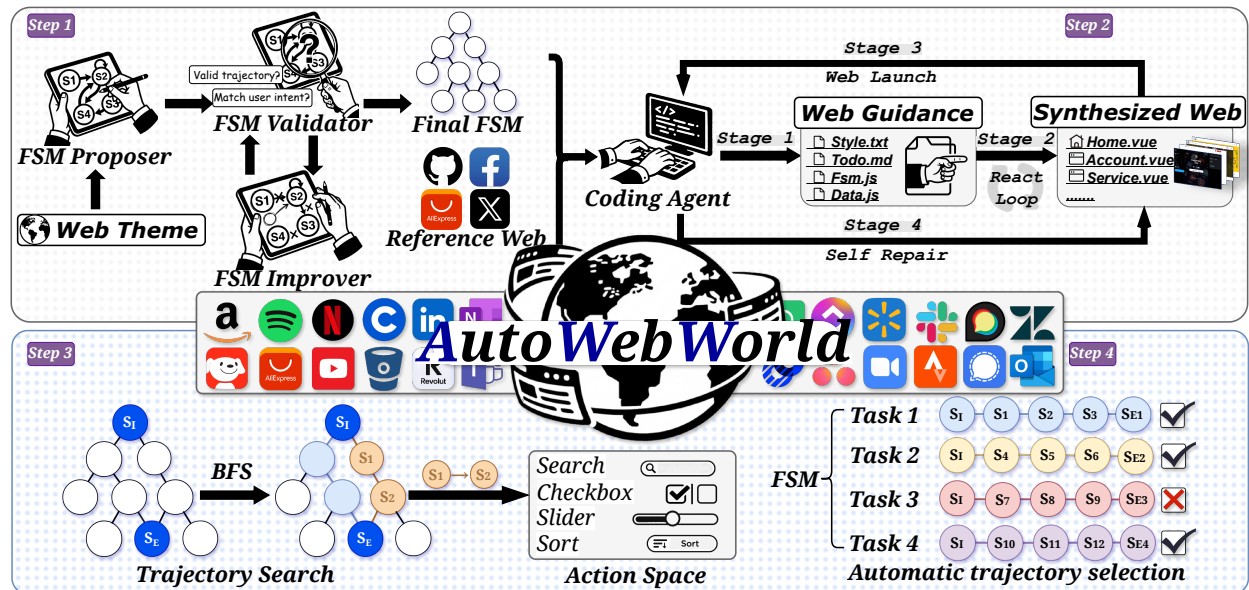

*Figure 2.* The four-step generation process of AutoWebWorld. Step 1 is to generate an FSM based on a multi-agent architecture. Step 2 uses coding agents to translate the output FSM into *Synthesized Web*. Step 3 uses BFS to explore the FSM graph and get all the potential trajectories. Step 4 filters these BFS-generated candidates by replaying each trajectory in the synthesized website with Playwright and retaining only those that execute all steps successfully and reach the intended goal state.

simulated websites, and they indicate that synthetic environments can provide diverse and stable verifiers that could also support data generation. Recent efforts start to couple environment generation with trajectory synthesis, but they still operate at a limited scale and do not explicitly model a fully known state-transition structure (Zhang et al., 2026b; Anonymous, 2025). AutoWebWorld pushes this direction further by adopting a transition-driven formulation and generating environments with intrinsic verification, enabling both reproducible benchmarking and batch data synthesis.

## 3. Preliminary

**GUI Trajectory.** We formalize Web GUI agent environment interaction as a Markov decision process $\mathcal{M} = \langle \mathcal{S}, \mathcal{A}, \mathcal{T}, \mathcal{O} \rangle$, where $\mathcal{S}$ is the environment's internal semantic state space, $\mathcal{A}$ is the executable action space, $\mathcal{T} : \mathcal{S} \times \mathcal{A} \to \mathcal{S}$ is a deterministic transition function, and $\mathcal{O}$ maps internal states to agent observations. At each time step $t$, the environment is at $s_t \in \mathcal{S}$ and emits an observation $o_t = \mathcal{O}(s_t)$; the agent selects an action $a_t \in \mathcal{A}$ based on $o_t$, and the environment updates to $s_{t+1} = \mathcal{T}(s_t, a_t)$. This induces an interaction trajectory $\tau = \{(o_t, a_t)\}_{t=1}^{T}$, where $o_t$ is typically a webpage screenshot with optional structured UI information. We explicitly distinguish the agent observation $o_t$ from the environment internal state $s_t$, since subsequent planning, trajectory synthesis, and automatic verification rely on deterministic state evolution rather than potentially ambiguous visual signals, which have been a common limitation in previous approaches that rely on raw screenshots to represent the state.

**State Definition.** To enable automatic trajectory enumeration over the state space, we define the environment's internal state as a pair of page identity and a page signature, $s_t = (p_t, \sigma_t)$, where $p_t \in \mathcal{P}$ denotes the current page id and $\sigma_t \in \Sigma_{p_t}$ is the signature of page $p_t$. More concretely, we represent the signature as an assignment of a set of structured variables that captures the page's controllable configuration and task context.

$$\sigma_t = \{ v_1^{(p_t)}, v_2^{(p_t)}, \dots, v_{K(p_t)}^{(p_t)} \}, \qquad v_k^{(p_t)} \in \mathcal{D}_k^{(p_t)}. \tag{1}$$

Here $v_k^{(p_t)}$ is the $k$-th signature variable on page $p_t$, and $\mathcal{D}_k^{(p_t)}$ is its domain. Different pages may involve different numbers and types of variables, so both $K(p_t)$ and the corresponding domains depend on the page. Intuitively, these variables cover key configurations that affect interaction outcomes and goal satisfaction, such as query text, filter and sorting options, pagination index, form field values, and selected item sets or cart contents. This representation distinguishes different interaction contexts and task progress within the same page, and provides a comparable and deduplicable semantic state basis for subsequent search on the state graph.

**State Transition.** We model the web interaction dynamics with a deterministic transition function

$$s_{t+1} = \mathcal{T}(s_t, a_t), \qquad \mathcal{T} : \mathcal{S} \times \mathcal{A} \to \mathcal{S}. \qquad (2)$$

Given the current state $s = (p, \sigma)$ and an action $a$, the environment first checks whether the action is applicable via a precondition predicate

$$\mathrm{pre}(s, a) \in \{0, 1\}. \qquad (3)$$

If $\mathrm{pre}(s, a) = 0$, the action is treated as invalid and handled by a fixed rule; the simplest choice is a no-op:

$$\mathcal{T}(s, a) = s. \qquad (4)$$

If $\mathrm{pre}(s, a) = 1$, the environment applies an effect rule $\mathrm{eff}(s, a)$ to update the signature and obtain a new signature $\sigma'$. We denote this deterministic update as $\mathrm{Apply}(\sigma, \mathrm{eff}(s, a))$. We distinguish two types of transitions. For an intra-page action, the page id remains unchanged, and only the signature is updated. For a navigation action, the page switches to a target page $p'$, whose signature is initialized by default and then merged with variables carried over from the previous context. Formally,

$$\mathcal{T}(s, a) = \begin{cases} \big(p, \ \mathrm{Apply}(\sigma, \mathrm{eff}(s, a))\big), & \text{in page,} \\ \big(p', \ \mathrm{Init}(p') \oplus \mathrm{Carry}(\sigma')\big), & \text{cross page.} \end{cases}$$
$$(5)$$

Here $\mathrm{Init}(p')$ initializes the default signature for page $p'$, $\mathrm{Carry}(\cdot)$ selects variables that should persist across pages, and $\oplus$ denotes a deterministic merge operator. With explicit preconditions and effect rules, the next state is uniquely determined by $(s, a)$, which enables systematic search over the state graph and automated trajectory verification and filtering.

## 4. AutoWebWorld

### 4.1. FSM Generation

As shown in Figure 2, the first step of AutoWebWorld is to generate an FSM. Based on the abstraction of state $s = (p, \sigma)$ and deterministic transition $\mathcal{T}$ in Section 3, we instantiate each website as an FSM specification for subsequent web generation and state-graph search. Given a web theme and reference website name, the goal of FSM generation is to produce a page set $\mathcal{P}$, define a signature space $\Sigma_p$ for each page, and specify executable transitions and goal states as a set of rules. Each transition is characterized by explicit precondition and effect rules, which ground executable interactions as a computable mapping $(s, a) \to s'$ and make goal attainment decidable from the internal state.

To improve FSM quality, we adopt a lightweight multi-agent generation and verification framework with a validator-driven loop. The FSM proposer first produces a candidate

FSM from the web theme, including the page set, per-page signature fields, and initial definitions of transitions and goal states. Detailed web theme used in AutoWebWorld are listed in Appendix C. The FSM validator then performs automatic checks on the candidate. If the validator determines that the FSM satisfies all requirements, such as ensuring that every terminal state has at least one reachable path and that the page transition logic aligns with real web behavior, it is directly returned as the final FSM. We show the detailed check requirements of the FSM validator in Appendix A.1. Otherwise, the validator outputs structured revision suggestions with localized issues, which are passed to the FSM improver. The improver revises the FSM accordingly and returns a completely new version for the validator to re-check. This *validate–revise* loop repeats until the validator accepts the FSM. The details of this pipeline for different agents are listed in Appendix A.1. All agents involved in this multi-agent framework, including the proposer, validator, and improver, are based on GPT-5.1 (OpenAI, 2025).

### 4.2. Web Environment Generation

The second step is to generate web environments based on the FSM. Given the final FSM from Section 4.1 and a reference website name, we generate an executable simulated web environment whose interactive behavior follows the FSM semantics. The reference website name serves as a style anchor, enabling diverse synthesized sites within the same theme category rather than replicating a single fixed design. We use coding agents, specifically Gemini3-Pro, to translate the FSM into a runnable front-end project through a 4-stage pipeline.

**Stage 1: Guidelines generation.** The coding agent first produces project-level guidelines and scaffolding that constrain subsequent code generation. These artifacts specify the visual style and component conventions, enumerate the required pages, and provide runtime representations needed remind the agent of the intended interaction logic. In our implementation, the outputs include a style specification, a task checklist, an executable FSM runtime module, and a mock data file that backs dynamic UI content.

**Stage 2: Web pages synthesis.** Conditioned on the Stage-1 guidelines, the agent generates page implementations as Vue components in batches. After producing a fixed number of pages, the agent performs a self-review by inspecting the current repository state and the guidelines, then updates the remaining to-do plan and implementation decisions before continuing. This generate–review–revise loop repeats until all pages and required interactions specified by the FSM are implemented.

**Stage 3: Web building.** After the page set is completed, we build and run the generated web project to check code-level executability. If the build succeeds, we finalize the

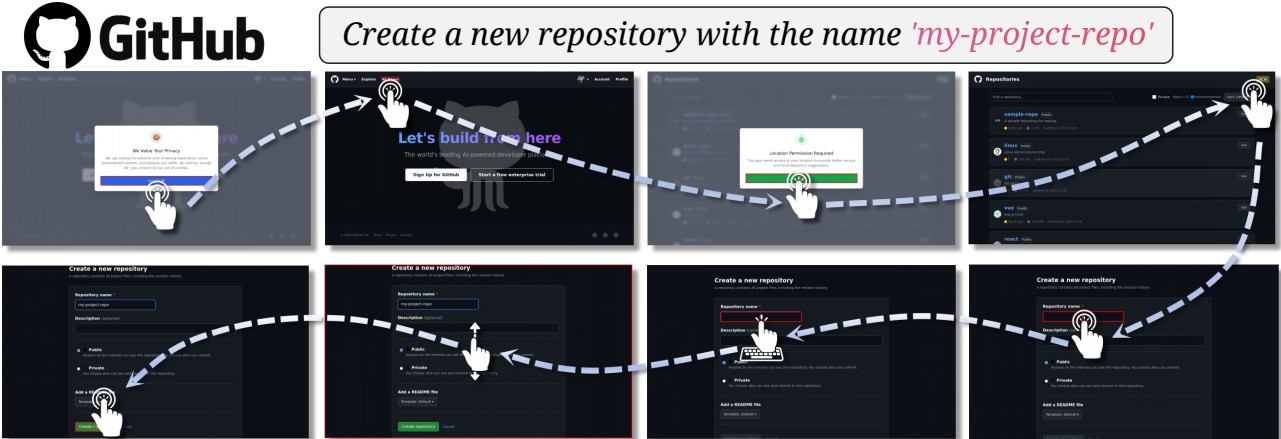

*Figure 3.* A verified, multi-step GUI trajectory for creating a repository in a synthesized GitHub environment. The action sequence is automatically generated by traversing the environment's underlying Finite State Machine (FSM) and programmatically verified through execution, demonstrating the end-to-end process described in Section 4.

environment.

**Stage 4: Self-repair.** If the build fails, we trigger a self-repair loop. The build logs and error messages are fed back to the coding agent, which modifies the project to fix the reported issues. We then rebuild and re-run the project. This repair loop repeats until the project builds successfully, producing a runnable simulated website that can be used for subsequent trajectory execution and filtering.

### 4.3. Automatic Trajectory Collection

Given the final FSM, we synthesize interaction trajectories by traversing its state graph with breadth-first search (BFS). Each node corresponds to an internal $s = (p, \sigma)$, and each edge corresponds to an executable transition induced by a high-level action $a$, namely $s' = \mathcal{T}(s, a)$. Importantly, each FSM action $a$ is accompanied by a pre-defined GUI procedure, *i.e.*, a short sequence of atomic UI operations (*e.g.*, click, type) that realizes the action on the rendered page. Starting from the initial state $s_0$, BFS explores the graph in layers and returns goal-reaching paths as action sequences. These paths are correct at the FSM level by construction and serve as candidates for trajectory synthesis. Crucially, the FSM actions are directly groundable to the generated front-end. During FSM specification, we assign a unique selector to every interactable UI component on each page. During web environment generation, the project strictly implements these selectors in the rendered DOM. This shared selector namespace provides a deterministic bridge from an FSM path to concrete UI operations. After BFS yields an action sequence, we expand it into the concatenated GUI procedures of all actions and replay the resulting atomic-operation sequence on the generated website using Playwright. At each procedure step, we locate the target element via its selector and obtain its bounding box, which instantiates the

corresponding atomic operation (*e.g.*, clicking at the box center) on the page. Executing the full sequence, therefore, simulates the corresponding user interaction, producing a grounded GUI trajectory aligned with the FSM path. These grounded trajectories form the collected dataset and are passed to the next stage for execution-based filtering. A trajectory example is shown in Figure 3.

### 4.4. Automatic Trajectory Filtering

Although trajectories obtained from BFS are valid at the FSM level and can be grounded via selectors, a small fraction of candidates may still fail when executed in the generated front-end due to website implementation mismatches. To address this, we perform execution-based filtering by batch replaying all collected trajectories on the web environment using Playwright. For each candidate trajectory, we sequentially execute its atomic-level GUI operations on the front-end. A trajectory is accepted only if all steps can be carried out successfully. This filtering rule is intentionally strict, as our goal is to obtain a reproducibly executable set of trajectories without relying on external judges. Specifically, during execution, if any action in the trajectory fails to match the expected element on the web page, the trajectory is discarded. Common problems include situations where the page fails to generate the element specified in the selector, or buttons or interactive elements do not function as expected, causing the trajectory to break. After filtering, the remaining trajectories form the final dataset used for training, with each trajectory paired with its grounded GUI actions and aligned internal state sequence. This process ensures that only valid, executable trajectories are included, improving the quality and consistency of the training data. After filtering the invalid trajectories, we generate queries based on the complete path with DeepSeek-V3.2.

*Table 1.* Comparison results between different GUI trajectory datasets.

| Datasets | # Website Pages | # Trajectories | Env Type | Verifier | Average Steps | Cost Per Trajectory |
|---|---|---|---|---|---|---|
| Explorer (Pahuja et al., 2025) | $49K$ | $94K$ | Real-World | External | 7.7 | $0.15 |
| AgentTerk (Xu et al., 2024) | 127 | 10,398 | Real-World | External | 12.1 | $0.55 |
| Fara (Awadallah et al., 2025) | 70,117 | 145,603 | Real-World | External | 6.9 | $1.00 |
| Mind2Web (Deng et al., 2023) | 137 | 2,350 | Real-World | External | 7.3 | $0.80 |
| **AutoWebWorld (Ours)** | **875** | **11,663** | **Synthesized** | **Inherent** | **21.9** | **$0.04** |

*Table 2.* Total cost for different stages.

| Stage | Models | Cost |
|---|---|---|
| FSM Generation | GPT-5.1 | $ 57.10 |
| Web Generation | Gemini3-Pro | $ 52.26 |
| Query Generation | DeepSeek-V3.2 | $ 65.84 |
| Thinking Generation | Gemini2.5-Flash | $ 272.17 |

### 4.5. Datasets Analysis

Table 1 compares representative GUI trajectory datasets in terms of website coverage, trajectory scale, environment type, verification mechanism, average horizon, and cost per trajectory. A clear trend is that real-world collections typically depend on external verifiers, so correctness is assessed outside the environment, and each additional trajectory incurs a non-trivial marginal cost. AutoWebWorld reduces the average cost to $0.04 per trajectory, compared with roughly $0.15–$1.00 in prior real-world pipelines. Table 2 further breaks down our generation cost and shows that the dominant expense comes from step-level thinking generation ($272.17), rather than environment construction or FSM generation.

AutoWebWorld also covers longer-horizon interactions. The average trajectory length is 21.94 steps, exceeding the 6.9–12.1 range of other datasets, suggesting stronger coverage of compositional behaviors that stress planning and state tracking. Although our total trajectory count is smaller than the largest real-world corpora, the combination of longer horizons and intrinsic verification yields reproducible, high-quality supervision, and the synthesized environments can be directly reused as stable GUI benchmarks with consistent success criteria.

## 5. Experiments

### 5.1. Experimental Setup

**Training Data.** We synthesize a total of 11,663 verified trajectories across all 29 generated websites. As shown in Figure 2, for each task, we generate multiple trajectories with different search strategies. Since these trajectories often share a high degree of similarity in their underlying transition sequences, we aim to reduce potential overfitting to homogeneous task patterns. To achieve this, we sample one trajectory from each set of parallel trajectories for the same task. This results in 1,215 distinct trajectories, containing a total of 12,585 interaction steps. In addition to trajectory rollouts, we further convert part of the trajectories into grounding supervision. Specifically, we extract individual steps from the trajectories and rewrite the corresponding queries to create grounding examples, enabling direct supervision for UI element localization. We then combine these grounding examples and trajectory steps into a unified training set. The training set, containing approximately 16k total training steps, is used for GRPO.

**Benchmarks.** To evaluate whether the synthesized data improves real-world generalization, we use WebVoyager (He et al., 2024) as the primary navigation benchmark. We further validate the scaling behavior of synthesized GUI trajectories on both WebVoyager (He et al., 2024) and Online-Mind2Web (Xue et al., 2025), testing whether performance improves predictably with increased training data. For grounding generalization, we evaluate on two widely used grounding benchmarks, ScreenSpot-V2 (Wu et al.) and ScreenSpot-Pro (Li et al., 2025). These benchmarks also allow us to test cross-resolution robustness, showing that our grounding data improves grounding accuracy across different screenshot resolutions. Finally, we use 25 AutoWebWorld trajectories as an additional benchmarking suite to evaluate existing SOTA GUI agents.

**Evaluation Details.** For WebVoyager, each agent is given the full textual history of previous thoughts and actions, while only the current screenshot is visible. We limit each task to 15 actions. In the WebVoyager and Online-Mind2Web, we use Gemini-3-Flash as the judge model to determine task success from the instruction and interaction trace. Since some websites frequently trigger access-denied errors or redirect to CAPTCHA mid-episode, we evaluate the 9 most stable websites and filter out the rest. To account for evaluation variance, we run each website on WebVoyager three times and report the mean across runs. To account for evaluation variance, we run each website on WebVoyager three times and report the mean across runs;

*Table 3.* Results on WebVoyager under 15 maximum steps by domain (%). CD represents the Cambridge Dictionary website. GF represents the Google Flights website. GM represents the Google Maps website. WA represents the Wolfram Alpha website. **Bold** means the first performance and underline means the second performance.

| Method | Apple | Arxiv | Coursera | CD | BBC | GF | GM | HuggingFace | WA | Overall |
|---|---|---|---|---|---|---|---|---|---|---|
| **Frontier Models** | | | | | | | | | | |
| GPT-5.1 | 28.12 | 4.76 | **50.0** | 23.26 | 12.50 | 0.0 | 0.0 | 8.11 | 39.13 | 18.96 |
| Claude-4.0-Sonnet | 34.38 | **26.19** | 20.0 | 39.53 | 28.12 | 0.0 | 0.0 | **35.14** | **47.83** | 26.11 |
| Gemini3-Pro | 9.38 | 0.0 | 7.5 | 9.3 | 6.25 | 0.0 | 10.0 | 10.81 | 2.17 | 5.46 |
| **Models ≤ 7B** | | | | | | | | | | |
| Qwen2.5-VL-3B | 6.25 | 4.76 | 0.0 | 11.63 | 0.0 | 0.0 | 0.0 | 0.0 | 13.04 | 3.96 |
| **Ours-3B** | 18.75 | 11.90 | 17.50 | 41.86 | 18.75 | 2.44 | 10.0 | 8.11 | 6.52 | 15.09 |
| **Models ≥ 7B** | | | | | | | | | | |
| Qwen2.5-VL-7B | 15.62 | 2.38 | 2.5 | 16.28 | 6.25 | 0.0 | 0.0 | 0.0 | 4.35 | 5.62 |
| Qwen2.5-VL-72B-Instruct | 12.50 | 2.38 | 15.00 | 39.53 | 3.12 | 0.0 | 7.5 | 8.11 | 17.39 | 11.73 |
| UI-Venus-7B | 9.38 | 7.14 | 17.50 | 16.28 | 15.62 | 0.0 | 0.0 | 10.81 | 10.87 | 9.73 |
| OpenCUA-7B | 15.62 | 7.14 | 17.5 | 60.47 | 3.12 | 2.44 | 10.0 | 18.92 | 10.87 | 16.74 |
| TongUI-7B | 18.75 | 7.14 | 17.50 | 6.98 | 18.75 | 0.0 | 2.5 | 8.11 | 8.70 | 9.83 |
| UI-TARS-1.5-7B | **34.38** | 23.81 | 27.50 | 46.51 | **31.25** | 0.0 | 10.0 | 21.62 | 43.48 | 26.51 |
| **Ours-7B** | 30.21 ± 4.78 | 19.84±2.75 | 27.50±2.50 | 68.99± 7.47 | 30.21±7.86 | 2.44±4.23 | 12.50±2.50 | 28.83±6.24 | 26.81±3.32 | **27.48**±0.35 |

the standard deviations reported in Table 3 are computed over these three runs. For scaling experiments on Online-Mind2Web, we keep the same setting and similarly remove websites with frequent access restrictions. For grounding, we follow the original evaluation protocols of ScreenSpot-V2 and ScreenSpot-Pro.

**Baseline Models.** We evaluate three groups of baselines spanning closed-source frontier models, open-source models below 7B, and larger than 7B. The closed source baselines include Gemini-3-Pro (Google, 2025), Claude-4-Sonnet (Anthropic, 2025), and GPT-5.1 (OpenAI, 2025). We evaluate all closed-source models via their official APIs. For open-source models, we include Qwen2.5-VL-3B (Bai et al., 2025), Qwen2.5-VL-7B (Bai et al., 2025), Qwen2.5-VL-72B-Instruct (Bai et al., 2025), UI-Venus-7B (Gu et al., 2025), OpenCUA-7B (Wang et al., 2025), TongUI-7B (Zhang et al., 2025a), and UI-TARS-1.5-7B (Qin et al., 2025).

### 5.2. Verified Trajectory for GUI Agent Training

We evaluate whether AutoWebWorld-synthesized trajectories improve GUI agents in two aspects. We first test real-world navigation generalization on WebVoyager. We then evaluate grounding on ScreenSpot-V2 and ScreenSpot-Pro to verify that the grounding supervision derived from AutoWebWorld also transfers.

#### 5.2.1. NAVIGATION RESULTS ON WEBVOYAGER

Table 3 shows that training on AutoWebWorld verified trajectories transfers to real-world navigation. Ours-7B achieves the best overall performance among open-source baselines on WebVoyager (27.42%), outperforming UI-TARS-1.5-7B (26.51%) and largely improving over Qwen2.5-VL-7B (5.62%). Gains are consistent across

domains, with strong results on CD (60.47%), Coursera (30.00%), and HuggingFace (32.43%), and non-trivial success on harder sites such as GF (7.32%) and GM (15.00%) where many models are near zero. Notably, Ours-3B reaches 15.09% overall, surpassing several 7B+ baselines, including UI-Venus-7B and TongUI-7B. This is particularly noteworthy because UI-Venus reports training with roughly 1M SFT samples and TongUI uses about 350K RL samples, whereas our results are obtained with only 16K synthesized steps for GRPO training, highlighting the data efficiency enabled by intrinsically verified synthetic trajectories.

#### 5.2.2. GROUNDING RESULTS ON SCREENSPOT-V2 AND SCREENSPOT-PRO

Table 5 and Table 6 show that AutoWebWorld-derived grounding data consistently improve performance on both ScreenSpot-V2 (Wu et al.) and ScreenSpot-Pro (Li et al., 2025). On ScreenSpot-V2, Ours-3B increases overall from 61.87 to 65.88 (+4.01), with notable gains on Desktop Text ($73.20 \rightarrow 80.41$) and Web Text/Icon ($63.68 \rightarrow 71.79$, $60.10 \rightarrow 66.50$). Ours-7B improves from 84.83 to 86.16 (+1.33), with the largest boosts on Desktop Icon ($65.71 \rightarrow 72.14$) and Desktop Text ($86.08 \rightarrow 89.69$). On ScreenSpot-Pro, improvements are larger and more consistent: Ours-3B rises from 13.3 to 18.0 (+4.7) and Ours-7B from 23.2 to 27.5 (+4.3), with strong gains on text grounding ($23.3 \rightarrow 30.9$ for 3B; $39.8 \rightarrow 46.1$ for 7B). Since the two benchmarks differ substantially in screenshot resolution—ScreenSpot-V2 spans standard mobile, desktop, and web interfaces while ScreenSpot-Pro targets high-resolution professional displays—the consistent gains on both indicate that the grounding supervision distilled from our synthesized environments transfers robustly across resolutions rather than overfitting to a particular screen scale.

*Table 4.* Comparison with external verification on WebVoyager under a matched training budget.

| METHOD | APPLE | ARXIV | COURSERA | CD | BBC | GF | GM | HF | WA | OVERALL |
|---|---|---|---|---|---|---|---|---|---|---|
| QWEN2.5-VL-7B | 15.62 | 2.38 | 2.50 | 16.28 | 6.25 | 0.00 | 0.00 | 0.00 | 4.35 | 5.26 |
| EXPLORER | 12.50 | 2.38 | 21.88 | 55.81 | 22.50 | 0.00 | 2.50 | 10.81 | **26.09** | 17.56 |
| **OURS** | **21.88** | **16.67** | **30.00** | **72.09** | **37.50** | 0.00 | **7.50** | **24.32** | 23.91 | **25.99** |

### 5.2.3. COMPARISON WITH EXTERNAL VERIFICATION

Table 3 compares datasets by static attributes, but not by a key question: under a matched training budget, does AutoWebWorld's intrinsic verification supervise better than the external verification of prior real-world pipelines? We test this against Explorer (Pahuja et al., 2025), whose trajectories are externally verified rather than certified within the environment.

**Setup.** Following Explorer's data collection protocol, we use GPT-4o to crawl 1,500 popular websites and collect 1,500 self-exploration trajectories (approximately 10K steps), at a reproduced cost of $0.12 per trajectory. For a fair comparison, we restrict AutoWebWorld to a matched budget of roughly 10K steps. Both datasets are then used to train Qwen2.5-VL-7B-Instruct under an identical GRPO pipeline, and we evaluate on WebVoyager under the same 15-step setting.

**Overall Results.** As shown in Table 4, AutoWebWorld attains 25.99% overall, surpassing Explorer by +8.43% under matched training steps, while costing substantially less to collect ($0.04 vs. $0.12 per trajectory). The advantage holds across the large majority of domains, with the largest gains on Arxiv, BBC, and HuggingFace; the only exception is Google Flights, where both models fail to complete any task. Since the two settings are matched on training steps and share the same backbone and optimization pipeline, the improvement is attributable to supervision quality rather than data volume. This directly supports our core motivation: intrinsically verified synthetic trajectories offer stronger and cleaner supervision than externally-verified real-world trajectories. Online exploration also incurs a hidden content-safety cost: crawling live websites frequently surfaces unsafe or inappropriate content that must be filtered out before training, adding overhead to real-world pipelines. Since AutoWebWorld synthesizes every page from a controlled FSM specification, its environments are free of such content by construction and require no safety filtering.

### 5.3. Scaled Performance with Scaled Data

Figure 4 shows a clear scaling trend of AutoWebWorld synthesized data on real-world benchmarks. Specifically, the four points in the figure correspond to training set sizes of 8, 256, 1,024, and 16,253 samples. As the training sample size increases, the overall success rate on WebVoyager mono-

*Table 5.* Comparison results on ScreenSpot-v2 between vanilla Qwen2.5-VL models and Qwen2.5-VL trained on our datasets.

| Models | Mobile | | Desktop | | Web | | Overall |
|---|---|---|---|---|---|---|---|
| | Text | Icon | Text | Icon | Text | Icon | |
| Qwen2.5-VL-3B | **73.45** | 50.71 | 73.20 | 38.57 | 63.68 | 60.10 | 61.87 |
| Ours-3B | 71.72 | **54.50** | **80.41** | **40.00** | **71.79** | **66.50** | **65.88** |
| Qwen2.5-VL-7B | 95.86 | 78.20 | 86.08 | 65.71 | 91.88 | **79.80** | 84.83 |
| Ours-7B | **96.21** | **79.15** | **89.69** | **72.14** | **92.31** | 78.33 | **86.16** |

*Table 6.* Comparison results on ScreenSpot-Pro between vanilla Qwen2.5-VL models and Qwen2.5-VL trained on our datasets.

| Models | Overall | | |
|---|---|---|---|
| | Icon | Text | Average |
| Qwen2.5-VL-3B | 3.3 | 23.3 | 13.3 |
| Ours-3B | **5.0** | **30.9** | **18.0** |
| Qwen2.5-VL-7B | 6.6 | 39.8 | 23.2 |
| Ours-7B | **8.8** | **46.1** | **27.5** |

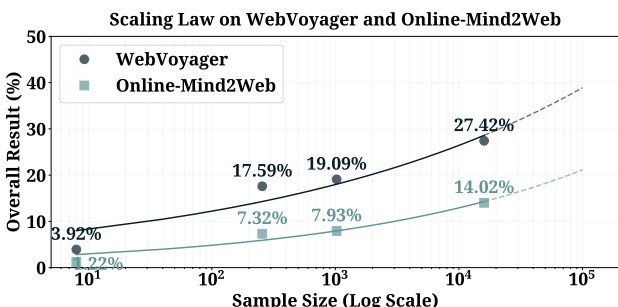

*Figure 4.* Scaling Curve on WebVoyager and Online-Mind2Web.

tonically improves from 3.92% → 17.59% → 19.09% → 27.42%, and Online-Mind2Web also increases from 1.22% → 7.32% → 7.93% → 14.02%. To characterize this trend, we perform a simple *polynomial fitting* on both curves. The fitted curves closely match the observed points and consistently predict further improvements in the higher-sample regime, suggesting that scaling up synthesized GUI trajectories yields sustained performance gains on real-world benchmarks. To ensure comparability across scales, we keep the training mixture fixed at every scale, maintaining a 2:8 ratio between grounding data and navigation (trajectory) data, so the curves primarily reflect the effect of increasing data volume rather than changes in data composition.

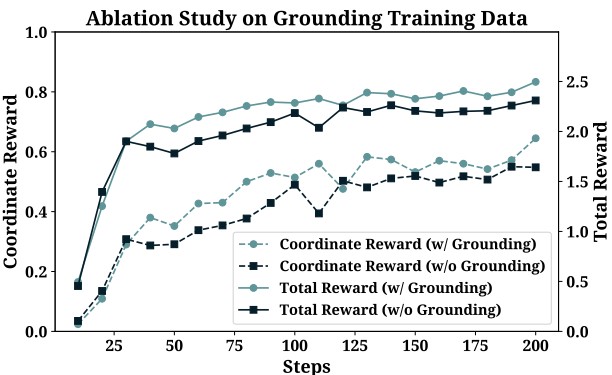

*Figure 5.* Ablation Study of Grounding Training Data on Qwen2.5-VL-3B.

*Table 7.* Comparison of frontier closed-source models on WebVoyager and AutoWebWorld.

| Env | Models | Success Rate |
|---|---|---|
| AutoWebWorld | UI-TARS-1.5-7B | 20.00 |
| | Claude-4-Sonnet | 16.00 |
| WebVoyager | UI-TARS-1.5-7B | 26.51 |
| | Claude-4-Sonnet | 26.11 |

### 5.4. Importance of Grounding Data in GUI Training

Figure 5 shows that grounding data plays a critical role in GRPO training with coordinate-based rewards. A notable pattern is that without grounding, the total reward is briefly higher at the very beginning. We attribute this to a difficulty shift: removing grounding samples excludes a subset of harder training instances, leading to a temporary reward advantage. However, this advantage quickly disappears. After roughly 25 steps, training with grounding exhibits a faster and more stable increase in the coordinate reward, and it remains consistently higher throughout the rest of the training, which in turn drives a sustained improvement in the total reward. In contrast, filtering out grounding data slows down the coordinate-reward growth and yields a lower plateau, ultimately limiting the achievable total reward. We also include all the GRPO reward functions in Appendix D.4.

### 5.5. Quality of Generated Environment

AutoWebWorld's synthesized websites can also naturally function as stable GUI benchmarks. To validate this benchmarking use case, we sample 25 trajectories from two AutoWebWorld websites cloned from Quora and GitHub, and evaluate two representative agents, UI-TARS-1.5-7B and Claude-4-Sonnet, on these tasks.

As shown in Table 7, although the success rates vary by model, each model achieves a lower trajectory success rate

on AutoWebWorld than on WebVoyager. This suggests that the synthesized websites are not trivially easier than real-world online evaluation; instead, they preserve comparable or stronger interaction challenges while remaining fully controllable and reproducibly testable.

## 6. Conclusion

We introduced AutoWebWorld, a transition-driven framework that synthesizes web environments from FSM specifications, which makes large-scale search-and-verify trajectory synthesis practical and low-cost, while also enabling controllable difficulty by predefining different FSMs with different page sets and transition structures. Empirically, we built 29 environments and synthesized 11,663 verified trajectories at $0.04 per trajectory; training with only 16k GRPO steps yields strong gains on real-world benchmarks and shows consistent improvement as synthetic data scales, suggesting verified synthetic web environments can effectively support generalization for foundational models.

## Limitations

**Realism bounded by generator priors.** Our environments are synthesized rather than crawled from the real web: an LLM proposes the FSM from a website name, and a coding agent renders it into a front-end. As a result, the interaction logic, edge cases, and UI conventions of each environment can only reach the generators' understanding of that website. Legitimate but uncommon interaction patterns that the models fail to anticipate are therefore absent from both the synthesized environments and the resulting training data, which may limit the diversity of behaviors an agent is exposed to.

**Semantic–GUI alignment gap.** Because correctness is verified at the FSM level while agents ultimately act on the rendered GUI, a fraction of FSM-valid trajectories are discarded when their actions cannot be reliably grounded due to front-end implementation mismatches. Consequently, certain FSM paths cannot be realized in execution, and coverage is implicitly traded for verifiability. We currently rely on strict execution-based filtering to guarantee correctness; introducing an automatic repair mechanism that reconciles such mismatches and recovers these trajectories is left to future work.

**Task diversity limited by FSM topology.** Since every task is instantiated as a goal-reaching path over the state graph and queries are generated from these paths, the space of expressible tasks is fixed by each FSM's page set and transition structure. Expanding task diversity therefore requires enriching the FSM specifications themselves, e.g., via larger page sets or more expressive transition rules.

## Impact Statement

AutoWebWorld reduces the cost and ambiguity of collecting web GUI trajectories by making task success verifiable inside the environment, rather than relying on external judges. This can accelerate research on reliable long-horizon web agents by enabling reproducible trajectory synthesis at scale, and by allowing systematic control of environment difficulty through FSM design. Beyond scalable training data, the same synthesized environments and verified trajectories can be directly packaged into stable GUI benchmarks with intrinsic success criteria, enabling consistent evaluation without brittle external checkers.

## Acknowledgments

This paper was supported by the NSF of China (62402409); Youth S&T Talent Support Programme of Guangdong Provincial Association for Science and Technology (SKXRC2025461); the Young Talent Support Project of Guangzhou Association for Science and Technology (QT-2025-001); Guangzhou Basic and Applied Basic Research Foundation (2026A1515010269, 2025A04J3935, 2023A1515110545); and Guangzhou-HKUST(GZ) Joint Funding Program (2025A03J3714).

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

# A. FSM Construction and Utilization

## A.1. FSM Construction

This subsection specifies the concrete `fsm.json` skeleton and its execution semantics, and describes how we construct a valid, verifiable FSM from a web theme. Unlike the high-level abstraction in the main text, we focus here on field-level definitions and constraints so that the FSM can be reproduced and executed directly.

### A.1.1. FILE LAYOUT AND GLOBAL META

Each environment is described by a single `fsm.json` file with four top-level components: `meta`, `pages`, `actions`, and `nav_skeleton`. The `meta` block declares the entry point and success terminals:

- `meta.initial_page_id` specifies the initial page.

- `meta.terminal_pages` lists the set of success terminal pages. The environment uses intrinsic criteria for success, where the simplest criterion is reaching a terminal page.

- `meta.complexity_profile` summarizes complexity and interceptor switches for generation/statistics only; it does not participate in transition semantics.

To ensure the intrinsic verifier is meaningful, every page in `terminal_pages` must be reachable from `initial_page_id` and correspond to an unambiguous goal-attainment outcome.

### A.1.2. STATE DEFINITION VIA PAGE ID AND SIGNATURE

We define an internal state as a pair $s = (p, \sigma)$, where $p$ is a discrete page id and $\sigma$ is the page *signature*. The signature is a structured assignment to task-relevant variables that makes otherwise implicit semantic state explicit. Concretely, each page declares its variables and default values in `pages[p].signature`. Downstream BFS enumeration and deduplication use $(p, \text{hash}(\sigma))$, so $\sigma$ must satisfy:

- **Minimality:** include only variables that affect task success or future reachability, avoiding UI noise that would unnecessarily inflate the state space.

- **Stability:** fixed variable names and hierarchical paths, deterministic default values, and a unique serialization order to make $\text{hash}(\sigma)$ reproducible.

In addition, we require every condition path to be a signature path that starts with `$.` (no placeholders), enforcing that verifiability comes from explicit state rather than implicit rendering cues.

### A.1.3. ACTION SCHEMA AND DETERMINISTIC TRANSITION SEMANTICS

FSM edges are defined in `actions`. Each action contains four semantic fields—`preconditions`, `effects`, `is_navigation`, and (if navigational) `to_page_id`—plus an execution field `gui_procedure`. We define a fixed, deterministic execution order:

**(1) Preconditions.** `preconditions` is a conjunction of boolean constraints evaluated only on the current signature. Each constraint must reference a signature path (starting with `$.`) and must not depend on screenshot-level element existence. This makes action availability verifiable and robust to rendering variation.

**(2) Effects.** If and only if all preconditions hold, we apply `effects` to update the current-page signature. Effects must be deterministic and satisfy a **local-update constraint**: an action may update only the fields it explicitly declares, leaving all other signature fields unchanged. We only allow enumerable, dependency-free update patterns such as assignment, counter increment/decrement, boolean toggle, enum switch, and set insert/delete.

**(3) Navigation.** If `is_navigation=true`, we navigate to `to_page_id` after applying effects. The target page signature is first initialized with that page's defaults, then deterministically merged with a carry-over subset of same-named fields. Navigation actions are allowed to have local effects (e.g., storing a selected item id) before page transition.

**(4) Result-set and pagination reset.** If an action changes the result set (e.g., search, filter, sort), pagination-related fields must be reset within `effects`, preventing semantic ambiguity where the result set changes but the page index remains stale.

With these rules, the successor state is uniquely determined by $(p, \sigma, a)$ whenever the action is executable, enabling step-wise verification by signature evolution.

### A.1.4. EXECUTABLE GROUNDING VIA `GUI_PROCEDURE`

`gui_procedure` expands a semantic action into a sequence of GUI-atomic operations (e.g., `scroll_until_visible`, `click`, `type_text`, and bounded page-next loops when needed). Importantly, `gui_procedure` specifies *how* to execute an action, but not *what* the semantic outcome is: semantic correctness is defined solely by `preconditions` and `effects` in signature space. For replay consistency and coordinate-level supervision, procedures use a normalized coordinate system in $[0, 1]^2$ and perform frame-consistency checks during execution.

### A.1.5. NAVIGATION SKELETON AS A DERIVED SUBGRAPH

`nav_skeleton` is a derived cross-page subgraph extracted from `actions` for structural visualization and reachability checks. It contains only cross-page edges, and each edge's `via` must reference an action with `is_navigation=true`. `nav_skeleton` does not introduce new transition semantics; the executable semantics are defined exclusively in `actions`.

### A.1.6. CONSTRUCTION PIPELINE AND VALIDITY CONSTRAINTS

We construct `fsm.json` with a multi-agent generate–validate–revise loop, which operationalizes the specification and constraints described above. Concretely, a *Proposer* agent drafts an initial FSM skeleton given a web theme and target task set, including (i) the page set and per-page signature defaults, (ii) the full action inventory with `preconditions`, `effects`, `is_navigation`/`to_page_id`, and (iii) `nav_skeleton` as a derived cross-page subgraph. Next, a *Validator* agent performs deterministic checks on the draft FSM.

These checks focus on properties required by our downstream BFS utilization: (1) all terminal pages in `meta.terminal_pages` are reachable from `meta.initial_page_id`; (2) each action's `preconditions` references only signature paths (starting with `$.`) and contains no placeholders; (3) `effects` are deterministic and satisfy the local-update constraint (only declared fields may change); (4) navigation transitions are well-defined, with deterministic target-page initialization and same-name merge; and (5) actions that modify result sets (search/filter/sort) explicitly reset pagination-related signature fields. If any check fails, an *Improver* agent revises only the implicated pages/actions while preserving previously validated components, and the Validator is rerun until all checks pass.

This multi-agent pipeline does not conflict with the field-level construction described earlier: the preceding subsections define *what* constitutes a valid, verifiable FSM (the schema and execution semantics), while the multi-agent loop defines *how* we reliably instantiate such FSMs at scale. In practice, the loop yields a stable `fsm.json` that is directly executable and supports intrinsic verification, which is essential for reproducible BFS-based trajectory synthesis and task-instance generation.

## A.2. BFS Mapping with FSM

This subsection describes how we utilize a constructed `fsm.json` to systematically enumerate reachable semantic states and synthesize verifiably correct trajectories and task instances. The key idea is to treat the FSM as an explicit, deterministic state-transition system over $(p, \sigma)$ and run breadth-first search (BFS) with signature-based deduplication, so that both reachability and success can be checked intrinsically in the environment.

### A.2.1. SEARCH GRAPH INDUCED BY THE FSM

Given `pages` and `actions`, we define a directed graph over semantic states. A node is a pair $s = (p, \sigma)$ where $p$ is a page id and $\sigma$ is the signature assignment for that page. An action $a$ induces a directed edge $s \rightarrow s'$ if and only if all `preconditions` of $a$ hold on $\sigma$. The successor state $s'$ is computed deterministically by applying `effects` and then, if `is_navigation=true`, switching to `to_page_id` with target-page signature initialization and deterministic same-name merge, as defined in A.1.

To ensure reproducible enumeration and avoid redundant exploration, BFS uses a deduplication key

$$\text{key}(s) = (p, \text{hash}(\sigma)),$$

where $\text{hash}(\sigma)$ is computed by a canonical serialization of the signature (fixed field order, fixed representation). A node is expanded at most once per key.

### A.2.2. BFS EXPANSION RULES AND PRACTICAL CONSTRAINTS

Starting from $s_0 = (p_0, \sigma_0)$ where $p_0 = \texttt{meta.initial\_page\_id}$ and $\sigma_0$ is the default signature of $p_0$, BFS expands the frontier in increasing path length. At a node $s$, we enumerate candidate actions available on page $p$ and keep only those with satisfied $\texttt{preconditions}$. Applying an action yields a unique successor state, so the BFS tree defines shortest paths in the semantic state space.

In practice, we impose two constraints to keep enumeration bounded and aligned with our downstream usage. First, we cap the maximum depth $L$ to control long-horizon blow-up, and we optionally cap the number of expanded nodes per page or per goal type to balance coverage. Second, we treat $\texttt{nav\_skeleton}$ purely as a structural aid: it provides the cross-page connectivity for sanity checks, while the actual expansion uses $\texttt{actions}$ so that both in-page actions and navigation actions are explored under the same semantic rules.

### A.2.3. MAPPING TASKS TO GOAL PREDICATES IN SIGNATURE SPACE

BFS is used to map task requirements into intrinsic goal conditions that can be verified without external judges. We represent each task as a goal predicate $G(s)$ defined on semantic states. The most direct form is reaching a success terminal page:

$$G(s) = \mathbb{I}[\, p \in \texttt{meta.terminal\_pages}\,].$$

More generally, tasks can be expressed as conjunctions of signature constraints on the terminal (or intermediate) state, for example "a specific item is selected" or "a form field is filled" or "a cart contains at least one item". Because all constraints reference explicit signature paths, $G(s)$ is deterministic and evaluation does not depend on screenshot-level heuristics.

During BFS, we check $G(s)$ upon dequeuing a node. When $G(s)$ holds, the path from $s_0$ to $s$ defines a verified solution trajectory, and BFS guarantees it is shortest among all trajectories reaching that goal key.

### A.2.4. TRAJECTORY EXTRACTION AND INTRINSIC VERIFICATION

Each BFS-discovered solution is stored as an action sequence $(a_1, \ldots, a_T)$ together with the corresponding semantic state sequence $(s_0, \ldots, s_T)$. Intrinsic verification is performed by replaying the semantic transitions: for every step $t$, we re-check that $\texttt{preconditions}$ hold at $s_{t-1}$ and that applying $\texttt{effects}$ (and navigation semantics if applicable) yields $s_t$. Final success is verified by $G(s_T)$, including terminal-page membership when used. Since transitions are deterministic and defined entirely in the FSM, verification has zero marginal human cost and is reproducible across runs.

To increase data diversity without sacrificing verifiability, we optionally sample multiple distinct trajectories that reach the same goal predicate but end in different semantic keys, for example by varying search queries, filters, or navigation choices. We also support generating negative trajectories by truncating before goal satisfaction or by forcing actions whose preconditions are unsatisfied (recorded as invalid), which can be used for training robustness, while keeping the verification signal explicit.

### A.2.5. LINKING SEMANTIC ACTIONS TO EXECUTABLE GUI PROCEDURES

For each semantic action in a BFS trajectory, we attach its $\texttt{gui\_procedure}$ to obtain an executable GUI-atomic sequence. This yields a two-level representation:

- a semantic-level trajectory in the FSM action space, used for state evolution and verification;

- a GUI-level trajectory composed of atomic operations, used for replay and for coordinate-level supervision.

Crucially, the semantic transition is the source of truth for correctness: the GUI procedure provides an execution realization, but correctness is determined by the FSM-defined state update and goal predicate. This separation enables us to scale trajectory synthesis while preserving deterministic verification.

### A.2.6. OUTPUTS OF BFS MAPPING

The BFS mapping produces (i) a set of verified task instances defined by goal predicates and initial conditions, and (ii) a set of verified trajectories with step-wise semantic states and grounded GUI procedures. These artifacts are directly used for large-scale training data synthesis and for benchmarking, since success and step validity are determined intrinsically by the FSM-defined transitions and goal attainment. Here is an example for `bfs.json`.

```json
{
  "trajectory": [
    {
      "id": "ACT_HOME_ACCEPT_COOKIES",
      "gui_procedure": [
        {
          "op": "click",
          "selector": "#cookie-accept"
        }
      ]
    },
    {
      "id": "ACT_HOME_OPEN_BASES_HOVER",
      "gui_procedure": [
        {
          "op": "hover",
          "selector": "#topbar-workspace-menu"
        },
        {
          "op": "click",
          "selector": "#topbar-workspace-menu .option-bases",
          "ui_elements": {
            "container": "#topbar-workspace-menu",
            "options": [
              {
                "value": "bases",
                "selector": "#topbar-workspace-menu .option-bases"
              },
              {
                "value": "workspaces",
                "selector": "#topbar-workspace-menu .option-workspaces"
              },
              {
                "value": "templates",
                "selector": "#topbar-workspace-menu .option-templates"
              }
            ]
          }
        }
      ]
    },
    {
      "id": "ACT_BASES_GRANT_LOCATION",
      "gui_procedure": [
        {
          "op": "click",
          "selector": "#permission-location-allow"
        }
      ]
    },
    {
      "id": "ACT_BASES_SORT",
      "gui_procedure": [
        {
          "op": "click",
          "selector": "#bases-sort-dropdown"
        },
        {
```

```
        "op": "click",
        "selector": "#bases-sort-recent-desc",
        "ui_elements": {
          "container": "#bases-sort-dropdown",
          "options": [
            {
              "value": "recent",
              "selector": "#bases-sort-recent-desc"
            },
            {
              "value": "alphabetical",
              "selector": "#bases-sort-alpha"
            },
            {
              "value": "starred",
              "selector": "#bases-sort-starred"
            }
          ]
        }
      }
    }
  ]
},
{
  "id": "ACT_BASES_OPEN_FILTERED_BASE",
  "gui_procedure": [
    {
      "op": "click",
      "selector": "#bases-grid .base-card-filtered"
    }
  ]
},
{
  "id": "ACT_BASE_WORKSPACE_OPEN_AUTOMATIONS",
  "gui_procedure": [
    {
      "op": "click",
      "selector": "#nav-automations"
    }
  ]
},
{
  "id": "ACT_AUTOMATIONS_OPEN_CREATE",
  "gui_procedure": [
    {
      "op": "click",
      "selector": "#create-automation-button"
    }
  ]
},
{
  "id": "ACT_AUTOMATION_SELECT_TRIGGER",
  "gui_procedure": [
    {
      "op": "click",
      "selector": "#trigger-dropdown"
    },
    {
      "op": "click",
      "selector": "#trigger-record-created",
      "ui_elements": {
        "container": "#trigger-dropdown",
        "options": [
          {
            "value": "when-record-created",
            "selector": "#trigger-record-created"
```

```
        },
        {
          "value": "when-record-updated",
          "selector": "#trigger-record-updated"
        },
        {
          "value": "at-scheduled-time",
          "selector": "#trigger-scheduled-time"
        }
      ]
    }
  }
]
},
{
  "id": "ACT_AUTOMATION_NEXT_TO_ACTION",
  "gui_procedure": [
    {
      "op": "click",
      "selector": "#next-to-action"
    }
  ]
},
{
  "id": "ACT_AUTOMATION_SELECT_ACTION",
  "gui_procedure": [
    {
      "op": "click",
      "selector": "#action-dropdown"
    },
    {
      "op": "click",
      "selector": "#action-send-email",
      "ui_elements": {
        "container": "#action-dropdown",
        "options": [
          {
            "value": "send-email",
            "selector": "#action-send-email"
          },
          {
            "value": "update-record",
            "selector": "#action-update-record"
          },
          {
            "value": "create-record",
            "selector": "#action-create-record"
          }
        ]
      }
    }
  ]
},
{
  "id": "ACT_AUTOMATION_SAVE",
  "gui_procedure": [
    {
      "op": "click",
      "selector": "#save-automation-button"
    }
  ]
}
]
}
```

## A.3. FSM Example

We present a compact `fsm.json` skeleton to illustrate how different components fit together in our specification. Concretely, `meta` defines global properties (entry page and success terminals), `pages` declares the page set with per-page signature defaults and the actions available on each page, `actions` specifies executable transitions with `preconditions`, deterministic `effects`, and grounded `gui_procedure`, and `nav_skeleton` provides a derived cross-page connectivity summary for visualization and reachability sanity checks. Since a full FSM for a realistic web app contains many pages and actions and is therefore lengthy, we only show an abbreviated skeleton here. The complete FSM specifications for all environments will be released in our open-source repository.

```
{
  "meta": {
    "app": "<APP_NAME>",
    "version": "1.0",
    "initial_page_id": "<PAGE_ID_HOME>",
    "terminal_pages": ["<PAGE_ID_SUCCESS_1>", "<PAGE_ID_SUCCESS_2>"],
    "complexity_profile": {
      "interceptors": {
        "cookie": true,
        "permissions": ["location"],
        "captcha": false,
        "register_correction": false
      },
      "pages": { "min": 20, "max": 35 },
      "actions": { "per_page_min": 6, "per_page_max": 8 },
      "terminals": { "count": 5 },
      "path_length": { "len": [6, 8], "shortest_only": true }
    }
  },

  "pages": {
    "<PAGE_ID_HOME>": {
      "page_name": "<HOME_PAGE_NAME>",
      "signature": {
        "<sig_field_1>": "<default_value_1>",
        "<sig_field_2>": "<default_value_2>",
        "pagination": { "page_index": 1 }
      },
      "actions": ["<ACT_ID_1>", "<ACT_ID_2>", "<ACT_ID_NAV_1>"]
    },

    "<PAGE_ID_LIST>": {
      "page_name": "<LIST_PAGE_NAME>",
      "signature": {
        "query": "",
        "filters": {},
        "sort_by": "<default_sort>",
        "pagination": { "page_index": 1 }
      },
      "actions": ["<ACT_ID_SEARCH>", "<ACT_ID_FILTER>", "<ACT_ID_SORT>", "<
  ACT_ID_OPEN_ITEM>"]
    },

    "<PAGE_ID_DETAIL>": {
      "page_name": "<DETAIL_PAGE_NAME>",
      "signature": {
        "selected_item_id": null,
        "<detail_field_1>": "<default_value>",
        "<detail_field_2>": "<default_value>"
      },
      "actions": ["<ACT_ID_ADD>", "<ACT_ID_BACK>", "<ACT_ID_NAV_NEXT>"]
    },

    "<PAGE_ID_SUCCESS_1>": {
```

```
      "page_name": "<SUCCESS_PAGE_NAME_1>",
      "signature": {},
      "actions": []
    }
  },

  "actions": {
    "<ACT_ID_1>": {
      "name": "<ACTION_NAME>",
      "from": "<PAGE_ID_HOME>",
      "to": "<PAGE_ID_HOME>",
      "is_navigation": false,
      "params": {},
      "preconditions": [
        { "path": "$.<sig_field_1>", "op": "==", "value": "<some_value>" }
      ],
      "effects": [
        { "path": "$.<sig_field_2>", "op": "assign", "value": "<new_value>" }
      ],
      "gui_procedure": [
        { "op": "click", "selector": "<CSS_SELECTOR>" }
      ]
    },

    "<ACT_ID_NAV_1>": {
      "name": "<NAV_ACTION_NAME>",
      "from": "<PAGE_ID_HOME>",
      "to": "<PAGE_ID_LIST>",
      "is_navigation": true,
      "to_page_id": "<PAGE_ID_LIST>",
      "params": {},
      "preconditions": [],
      "effects": [],
      "gui_procedure": [
        { "op": "click", "selector": "<CSS_SELECTOR_NAV>" }
      ]
    },

    "<ACT_ID_SEARCH>": {
      "name": "search",
      "from": "<PAGE_ID_LIST>",
      "to": "<PAGE_ID_LIST>",
      "is_navigation": false,
      "params": { "query": "<QUERY_PLACEHOLDER>" },
      "preconditions": [],
      "effects": [
        { "path": "$.query", "op": "assign", "value": "<QUERY_PLACEHOLDER>" },
        { "path": "$.pagination.page_index", "op": "assign", "value": 1 }
      ],
      "gui_procedure": [
        { "op": "click", "selector": "<SEARCH_BOX_SELECTOR>" },
        { "op": "type_text", "text": "<QUERY_PLACEHOLDER>" },
        { "op": "click", "selector": "<SEARCH_SUBMIT_SELECTOR>" }
      ]
    },

    "<ACT_ID_SORT>": {
      "name": "select",
      "from": "<PAGE_ID_LIST>",
      "to": "<PAGE_ID_LIST>",
      "is_navigation": false,
      "params": { "widget": "sort" },
      "preconditions": [],
      "effects": [
        { "path": "$.sort_by", "op": "assign", "value": "<SORT_OPTION>" },
```

```
            { "path": "$.pagination.page_index", "op": "assign", "value": 1 }
          ],
          "gui_procedure": [
            { "op": "click", "selector": "<SORT_DROPDOWN_SELECTOR>" },
            {
              "op": "click",
              "selector": "<SORT_OPTION_SELECTOR>",
              "ui_elements": {
                "container": "<SORT_DROPDOWN_SELECTOR>",
                "options": [
                  { "value": "<SORT_OPTION_1>", "selector": "<SORT_OPTION_1_SELECTOR>" },
                  { "value": "<SORT_OPTION_2>", "selector": "<SORT_OPTION_2_SELECTOR>" }
                ]
              }
            }
          ]
        },

      "<ACT_ID_OPEN_ITEM>": {
        "name": "click",
        "from": "<PAGE_ID_LIST>",
        "to": "<PAGE_ID_DETAIL>",
        "is_navigation": true,
        "to_page_id": "<PAGE_ID_DETAIL>",
        "params": { "item_id": "<ITEM_ID_PLACEHOLDER>" },
        "preconditions": [],
        "effects": [
          { "path": "$.selected_item_id", "op": "assign", "value": "<ITEM_ID_PLACEHOLDER>" }
        ],
        "gui_procedure": [
          { "op": "click", "selector": "<ITEM_CARD_SELECTOR>" }
        ]
      }
    }
  },

  "nav_skeleton": {
    "nodes": ["<PAGE_ID_HOME>", "<PAGE_ID_LIST>", "<PAGE_ID_DETAIL>", "<PAGE_ID_SUCCESS_1>
    "],
    "edges": [
      { "from": "<PAGE_ID_HOME>", "to": "<PAGE_ID_LIST>", "via": "<ACT_ID_NAV_1>" },
      { "from": "<PAGE_ID_LIST>", "to": "<PAGE_ID_DETAIL>", "via": "<ACT_ID_OPEN_ITEM>" },
      { "from": "<PAGE_ID_DETAIL>", "to": "<PAGE_ID_SUCCESS_1>", "via": "<ACT_ID_NAV_NEXT>
      " }
    ]
  }
}
```

## B. Datasets Construction Details and Statistical Analysis

This subsection describes the end-to-end pipeline that converts each synthesized website into a collection of query instances with intrinsic ground truth, together with the artifacts used for reproducible statistical analysis. For every environment, we assume three environment-side files are available: (i) fsm.json, which defines executable semantic transitions and grounded GUI procedures. We have already given an example in A.3 ; (ii) bfs.json, which enumerates BFS-derived trajectories over the FSM action graph. We have already given an example in A.2.6; and (iii) data.js, a pre-specified structured data file that contains the underlying feature values for all items rendered by the website. In addition, we generate one feature-conditioned image per item based on data.js, which enables queries grounded in visual descriptions and screenshot content. Here we show an example of data.js:

```
import { defineStore } from 'pinia'

export const useDataStore = defineStore('data', {
  state: () => ({
    // PROVIDERS
```

```
providers: [
  { id: 'prov_1', name: 'Dr. Sarah Johnson', specialty: 'Primary Care', rating: 4.9,
image: '/images/providers_prov_1.jpg', next_slot: 'Today' },
  { id: 'prov_2', name: 'Dr. Michael Chen', specialty: 'Cardiology', rating: 4.8,
image: '/images/providers_prov_2.jpg', next_slot: 'Tomorrow' },
  { id: 'prov_3', name: 'Dr. Emily Davis', specialty: 'Dermatology', rating: 4.7,
image: '/images/providers_prov_3.jpg', next_slot: 'Today' },
  { id: 'prov_4', name: 'Dr. Robert Wilson', specialty: 'Primary Care', rating: 4.5,
image: '/images/providers_prov_4.jpg', next_slot: 'In 2 days' },
  { id: 'prov_5', name: 'Dr. Jessica Taylor', specialty: 'Pediatrics', rating: 4.9,
image: '/images/providers_prov_5.jpg', next_slot: 'Today' },
  { id: 'prov_6', name: 'Dr. David Anderson', specialty: 'Primary Care', rating: 4.6,
image: '/images/providers_prov_6.jpg', next_slot: 'Tomorrow' },
  { id: 'prov_7', name: 'Dr. Jennifer Martinez', specialty: 'Dermatology', rating:
4.8, image: '/images/providers_prov_7.jpg', next_slot: 'In 3 days' },
  { id: 'prov_8', name: 'Dr. James Thomas', specialty: 'Cardiology', rating: 4.7,
image: '/images/providers_prov_8.jpg', next_slot: 'Today' },
  { id: 'prov_9', name: 'Dr. Lisa White', specialty: 'Primary Care', rating: 4.4,
image: '/images/providers_prov_9.jpg', next_slot: 'Tomorrow' },
  { id: 'prov_10', name: 'Dr. Daniel Harris', specialty: 'Orthopedics', rating: 4.9,
image: '/images/providers_prov_10.jpg', next_slot: 'In 2 days' },
  { id: 'prov_11', name: 'Dr. Mary Martin', specialty: 'Primary Care', rating: 4.8,
image: '/images/providers_prov_11.jpg', next_slot: 'Today' },
  { id: 'prov_12', name: 'Dr. Christopher Thompson', specialty: 'Neurology', rating:
4.7, image: '/images/providers_prov_12.jpg', next_slot: 'Tomorrow' },
  { id: 'prov_13', name: 'Dr. Patricia Garcia', specialty: 'Dermatology', rating: 4.6,
 image: '/images/providers_prov_13.jpg', next_slot: 'In 4 days' },
  { id: 'prov_14', name: 'Dr. Matthew Robinson', specialty: 'Primary Care', rating:
4.9, image: '/images/providers_prov_14.jpg', next_slot: 'Today' },
  { id: 'prov_15', name: 'Dr. Elizabeth Clark', specialty: 'Pediatrics', rating: 4.8,
image: '/images/providers_prov_15.jpg', next_slot: 'Tomorrow' },
  { id: 'prov_16', name: 'Dr. Joseph Rodriguez', specialty: 'Cardiology', rating: 4.7,
 image: '/images/providers_prov_16.jpg', next_slot: 'In 2 days' },
  { id: 'prov_17', name: 'Dr. Linda Lewis', specialty: 'Primary Care', rating: 4.5,
image: '/images/providers_prov_17.jpg', next_slot: 'Today' },
  { id: 'prov_18', name: 'Dr. Thomas Lee', specialty: 'Orthopedics', rating: 4.8,
image: '/images/providers_prov_18.jpg', next_slot: 'Tomorrow' },
  { id: 'prov_19', name: 'Dr. Barbara Walker', specialty: 'Dermatology', rating: 4.6,
image: '/images/providers_prov_19.jpg', next_slot: 'In 3 days' },
  { id: 'prov_20', name: 'Dr. Charles Hall', specialty: 'Primary Care', rating: 4.9,
image: '/images/providers_prov_20.jpg', next_slot: 'Today' }
],

// MENTAL HEALTH THERAPISTS
therapists: [
  { id: 'th_1', name: 'Amanda Wilson, LMFT', specialty: 'Anxiety & Depression',
experience: 10, image: '/images/therapists_th_1.jpg' },
  { id: 'th_2', name: 'Dr. Brian Miller, PsyD', specialty: 'Trauma', experience: 15,
image: '/images/therapists_th_2.jpg' },
  { id: 'th_3', name: 'Catherine Moore, LCSW', specialty: 'Family Therapy', experience
: 8, image: '/images/therapists_th_3.jpg' },
  { id: 'th_4', name: 'David Brown, LPC', specialty: 'Addiction', experience: 12,
image: '/images/therapists_th_4.jpg' },
  { id: 'th_5', name: 'Dr. Eleanor Davis, PhD', specialty: 'Child Psychology',
experience: 20, image: '/images/therapists_th_5.jpg' },
  { id: 'th_6', name: 'Frank Wright, LMFT', specialty: 'Couples Counseling',
experience: 14, image: '/images/therapists_th_6.jpg' },
  { id: 'th_7', name: 'Grace Green, LCSW', specialty: 'Anxiety', experience: 6, image:
 '/images/therapists_th_7.jpg' },
  { id: 'th_8', name: 'Dr. Henry Baker, PsyD', specialty: 'Depression', experience:
18, image: '/images/therapists_th_8.jpg' },
  { id: 'th_9', name: 'Isabella King, LPC', specialty: 'Stress Management', experience
: 9, image: '/images/therapists_th_9.jpg' },
  { id: 'th_10', name: 'Jack Scott, LMFT', specialty: 'Grief Counseling', experience:
11, image: '/images/therapists_th_10.jpg' },
```

```
    { id: 'th_11', name: 'Kelly Adams, LCSW', specialty: 'Anxiety & Depression',
experience: 7, image: '/images/therapists_th_11.jpg' },
    { id: 'th_12', name: 'Dr. Larry Nelson, PhD', specialty: 'Trauma', experience: 22,
image: '/images/therapists_th_12.jpg' },
    { id: 'th_13', name: 'Megan Carter, LPC', specialty: 'Family Therapy', experience:
5, image: '/images/therapists_th_13.jpg' },
    { id: 'th_14', name: 'Nathan Mitchell, LMFT', specialty: 'Addiction', experience:
13, image: '/images/therapists_th_14.jpg' },
    { id: 'th_15', name: 'Olivia Perez, LCSW', specialty: 'Child Psychology', experience
: 16, image: '/images/therapists_th_15.jpg' }
],

// PRESCRIPTIONS
prescriptions: [
    { id: 'rx_1', name: 'Lisinopril', dosage: '10mg', status: 'Active', supply: '30 days
', image: '/images/prescriptions_rx_1.jpg' },
    { id: 'rx_2', name: 'Metformin', dosage: '500mg', status: 'Active', supply: '90 days
', image: '/images/prescriptions_rx_2.jpg' },
    { id: 'rx_3', name: 'Atorvastatin', dosage: '20mg', status: 'Active', supply: '30
days', image: '/images/prescriptions_rx_3.jpg' },
    { id: 'rx_4', name: 'Levothyroxine', dosage: '50mcg', status: 'Active', supply: '90
days', image: '/images/prescriptions_rx_4.jpg' },
    { id: 'rx_5', name: 'Amlodipine', dosage: '5mg', status: 'Active', supply: '30 days'
, image: '/images/prescriptions_rx_5.jpg' },
    { id: 'rx_6', name: 'Metoprolol', dosage: '25mg', status: 'Inactive', supply: '30
days', image: '/images/prescriptions_rx_6.jpg' },
    { id: 'rx_7', name: 'Omeprazole', dosage: '20mg', status: 'Active', supply: '30 days
', image: '/images/prescriptions_rx_7.jpg' },
    { id: 'rx_8', name: 'Losartan', dosage: '50mg', status: 'Active', supply: '90 days',
 image: '/images/prescriptions_rx_8.jpg' },
    { id: 'rx_9', name: 'Gabapentin', dosage: '300mg', status: 'Active', supply: '30
days', image: '/images/prescriptions_rx_9.jpg' },
    { id: 'rx_10', name: 'Hydrochlorothiazide', dosage: '12.5mg', status: 'Inactive',
supply: '30 days', image: '/images/prescriptions_rx_10.jpg' },
    { id: 'rx_11', name: 'Sertraline', dosage: '50mg', status: 'Active', supply: '30
days', image: '/images/prescriptions_rx_11.jpg' },
    { id: 'rx_12', name: 'Simvastatin', dosage: '20mg', status: 'Active', supply: '90
days', image: '/images/prescriptions_rx_12.jpg' },
    { id: 'rx_13', name: 'Montelukast', dosage: '10mg', status: 'Active', supply: '30
days', image: '/images/prescriptions_rx_13.jpg' },
    { id: 'rx_14', name: 'Escitalopram', dosage: '10mg', status: 'Inactive', supply: '30
 days', image: '/images/prescriptions_rx_14.jpg' },
    { id: 'rx_15', name: 'Albuterol', dosage: '90mcg', status: 'Active', supply: '
Inhaler', image: '/images/prescriptions_rx_15.jpg' }
],

// APPOINTMENTS
appointments: [
    { id: 'apt_1', provider: 'Dr. Sarah Johnson', date: '2025-10-25', time: '10:00 AM',
type: 'Virtual', status: 'Upcoming', image: '/images/appointments_apt_1.jpg' },
    { id: 'apt_2', provider: 'Dr. Michael Chen', date: '2025-10-28', time: '02:30 PM',
type: 'In-Person', status: 'Upcoming', image: '/images/appointments_apt_2.jpg' },
    { id: 'apt_3', provider: 'Amanda Wilson, LMFT', date: '2025-11-01', time: '11:00 AM'
, type: 'Virtual', status: 'Upcoming', image: '/images/appointments_apt_3.jpg' },
    { id: 'apt_4', provider: 'Dr. Emily Davis', date: '2025-11-05', time: '09:15 AM',
type: 'Virtual', status: 'Upcoming', image: '/images/appointments_apt_4.jpg' },
    { id: 'apt_5', provider: 'Dr. Robert Wilson', date: '2025-11-10', time: '03:45 PM',
type: 'In-Person', status: 'Upcoming', image: '/images/appointments_apt_5.jpg' },
    { id: 'apt_6', provider: 'Dr. Sarah Johnson', date: '2025-09-15', time: '10:00 AM',
type: 'Virtual', status: 'Past', image: '/images/appointments_apt_6.jpg' },
    { id: 'apt_7', provider: 'Dr. Jessica Taylor', date: '2025-08-20', time: '01:00 PM',
 type: 'Virtual', status: 'Past', image: '/images/appointments_apt_7.jpg' },
    { id: 'apt_8', provider: 'Dr. David Anderson', date: '2025-07-10', time: '11:30 AM',
 type: 'In-Person', status: 'Past', image: '/images/appointments_apt_8.jpg' },
```

```javascript
      { id: 'apt_9', provider: 'Catherine Moore, LCSW', date: '2025-06-05', time: '04:00
   PM', type: 'Virtual', status: 'Past', image: '/images/appointments_apt_9.jpg' },
      { id: 'apt_10', provider: 'Dr. Jennifer Martinez', date: '2025-05-12', time: '09:00
   AM', type: 'Virtual', status: 'Past', image: '/images/appointments_apt_10.jpg' },
      { id: 'apt_11', provider: 'Dr. James Thomas', date: '2025-11-15', time: '02:00 PM',
   type: 'In-Person', status: 'Upcoming', image: '/images/appointments_apt_11.jpg' },
      { id: 'apt_12', provider: 'Dr. Lisa White', date: '2025-11-20', time: '10:30 AM',
   type: 'Virtual', status: 'Upcoming', image: '/images/appointments_apt_12.jpg' },
      { id: 'apt_13', provider: 'Dr. Daniel Harris', date: '2025-11-25', time: '01:45 PM',
    type: 'Virtual', status: 'Upcoming', image: '/images/appointments_apt_13.jpg' },
      { id: 'apt_14', provider: 'Dr. Mary Martin', date: '2025-12-01', time: '11:15 AM',
   type: 'In-Person', status: 'Upcoming', image: '/images/appointments_apt_14.jpg' },
      { id: 'apt_15', provider: 'Dr. Christopher Thompson', date: '2025-12-05', time: '
   03:30 PM', type: 'Virtual', status: 'Upcoming', image: '/images/appointments_apt_15.
   jpg' }
   ],

   // BILLS
   bills: [
      { id: 'bill_1', description: 'Office Visit - Dr. Johnson', date: '2025-09-15',
   amount: 50.00, status: 'Due', image: '/images/bills_bill_1.jpg' },
      { id: 'bill_2', description: 'Lab Work', date: '2025-09-15', amount: 25.00, status:
   'Due', image: '/images/bills_bill_2.jpg' },
      { id: 'bill_3', description: 'Therapy Session', date: '2025-08-20', amount: 80.00,
   status: 'Paid', image: '/images/bills_bill_3.jpg' },
      { id: 'bill_4', description: 'Specialist Consultation', date: '2025-07-10', amount:
   120.00, status: 'Paid', image: '/images/bills_bill_4.jpg' },
      { id: 'bill_5', description: 'X-Ray Services', date: '2025-06-05', amount: 150.00,
   status: 'Paid', image: '/images/bills_bill_5.jpg' },
      { id: 'bill_6', description: 'Annual Checkup', date: '2025-05-12', amount: 0.00,
   status: 'Paid', image: '/images/bills_bill_6.jpg' },
      { id: 'bill_7', description: 'Prescription Copay', date: '2025-09-10', amount:
   15.00, status: 'Due', image: '/images/bills_bill_7.jpg' },
      { id: 'bill_8', description: 'Telehealth Visit', date: '2025-08-15', amount: 40.00,
   status: 'Paid', image: '/images/bills_bill_8.jpg' },
      { id: 'bill_9', description: 'Blood Test', date: '2025-07-20', amount: 30.00, status
   : 'Paid', image: '/images/bills_bill_9.jpg' },
      { id: 'bill_10', description: 'MRI Scan', date: '2025-04-10', amount: 250.00, status
   : 'Paid', image: '/images/bills_bill_10.jpg' }
   ],

   // PLANS
   plans: [
      { id: 'plan_1', name: 'Gold Premium Plan', type: 'PPO', eligible: true, image: '/
   images/plans_plan_1.jpg' },
      { id: 'plan_2', name: 'Silver Saver Plan', type: 'HMO', eligible: true, image: '/
   images/plans_plan_2.jpg' },
      { id: 'plan_3', name: 'Bronze Basic Plan', type: 'EPO', eligible: true, image: '/
   images/plans_plan_3.jpg' },
      { id: 'plan_4', name: 'Dental Plus', type: 'Dental', eligible: false, image: '/
   images/plans_plan_4.jpg' },
      { id: 'plan_5', name: 'Vision Basic', type: 'Vision', eligible: true, image: '/
   images/plans_plan_5.jpg' }
   ]
  }),
  persist: {
   storage: sessionStorage
  }
})
```

Given these inputs, we synthesize three families of queries. The first family (BFS-driven) is generated directly from `bfs.json` by instantiating interaction templates over BFS-reachable trajectories. The second family (visual-grounded) is generated from `data.js` and the feature-conditioned item images, where the target item is referenced by a visual description rather than its name, while the interaction templates remain aligned with the BFS-driven setting. The third family

(screenshot QA) is generated by first sampling feature-based QA templates from `data.js`, then applying an additional VLM-based visibility check to ensure that the asked feature is actually present in the corresponding page screenshot; only verified-visible QA instances are kept.

For the first two families, we standardize five interaction modes that define how a target item is located or constrained: `search`, `scroll`, `slider`, `sort`, and `checkbox`. These modes share the same high-level template structure across families, but differ in the grounding signal used to identify the target item: name-based grounding for BFS-driven queries and visual-description grounding for visual-grounded queries. The screenshot QA family keeps the same QA template form as the underlying feature-based generator, but filters instances by screenshot visibility.

The pipeline outputs a set of per-environment query files (e.g., JSON/JSONL) grouped by family, as well as a lightweight statistics manifest that records counts and key attributes needed for analysis (family label, interaction mode, template parameters such as $n$ in "the $n$-th item", threshold values for sliders, sort keys, and checkbox conditions). All reported dataset statistics in this paper are computed solely from these released query files and manifests, ensuring full reproducibility.

## C. Cost Analysis

In this appendix, we report a fine-grained cost breakdown of the AutoWebWorld generation pipeline in Table 8, aggregated by category and individual websites. Specifically, we decompose the overall expense into four components: (i) **Web**, the cost of running coding agents to generate runnable websites; (ii) **FSM**, the cost of using a multi-agent workflow to produce the FSM specification; (iii) **Queries**, the cost of invoking advanced language models to generate queries from the filtered trajectories; and (iv) **Thinking**, the cost of generating step-level reasoning traces. Overall, Thinking dominates the total budget by a large margin, suggesting that the main bottleneck in long-horizon web tasks lies not in environment execution but in per-step model reasoning and planning. We also observe substantial variation across websites: Skyscanner, Walmart, Github, and Headspace incur markedly higher Thinking cost, consistent with longer interaction horizons and more complex decision-making requirements. In contrast, Web and FSM costs are relatively stable and much smaller in magnitude, reflecting largely fixed overheads for execution and deterministic verification. Beyond supporting reproducibility and budgeting, this breakdown points to a clear optimization direction: reducing redundant reasoning and unnecessary steps (e.g., via stronger action constraints, improved state visibility, and caching) is likely the most effective way to lower total cost without sacrificing task success.

## D. Model Implementation and Training Details

This section provides implementation and training details that are omitted from the main text due to space constraints, enabling faithful reproduction and a clear understanding of our engineering choices. Specifically, we summarize the software and hardware environment, define the unified low-level GUI action space, and (in the subsequent sections) report the full hyperparameter configurations and the reward functions used for GRPO, bridging the gap between the method description and an executable training pipeline.

### D.1. Software Environment

All training and inference experiments were conducted on a single machine equipped with 8× NVIDIA A800 GPUs. We implement the models and training pipeline in a standard PyTorch-based stack, and run distributed training across the 8 GPUs.

### D.2. GUI Action Space

### D.3. Hyperparameter Settings

We train our policy with GRPO using distributed data-parallel training on a single node with 8 GPUs (`torchrun`, `--nproc_per_node=8`). We initialize from a Qwen2.5-VL-7B checkpoint. The maximum prompt length is set to 1024 tokens (`--max_prompt_length 1024`). We use a per-device batch size of 4 (`--per_device_train_batch_size 4`) with 8 gradient accumulation steps (`--gradient_accumulation_steps 8`), resulting in an effective batch size of $4 \times 8 \times 8 = 256$ samples per optimization step. We enable BF16 mixed precision (`--bf16`) and gradient checkpointing (`--gradient_checkpointing true`) to reduce memory usage, and adopt FlashAttention-2 as the attention implementation (`--attn_implementation flash_attention_2`). For vision inputs, we cap the maximum pixels per

*Table 8.* Total cost for constructing AutoWebWorld.

| Category | Web | # Trajectories | Cost | | | | Total Cost |
|---|---|---|---|---|---|---|---|
| | | | **Web** | **FSM** | **Queries** | **Thinking** | |
| **Travelling** | Skyscanner | 1300 | $2.98 | $1.67 | $4.30 | $53.25 | $62.20 |
| **Commerce** | Aliexpress | 299 | $2.47 | $1.75 | $1.89 | $7.03 | $13.14 |
| | JD | 366 | $2.81 | $1.61 | $2.17 | $4.83 | $11.42 |
| | Revolut | 158 | $1.95 | $1.62 | $3.24 | $2.23 | $9.04 |
| | Shopee | 162 | $1.18 | $1.75 | $2.03 | $2.11 | $7.07 |
| | Shopify | 639 | $2.88 | $3.45 | $2.99 | $18.18 | $27.50 |
| | Walmart | 869 | $2.67 | $1.64 | $4.77 | $37.92 | $47.00 |
| **Productivity** | Airtable | 136 | $2.55 | $1.53 | $2.05 | $2.62 | $8.75 |
| | Asana | 273 | $1.01 | $1.44 | $5.82 | $3.62 | $11.89 |
| | Bitbucket | 285 | $2.21 | $3.68 | $2.74 | $3.93 | $12.56 |
| | FreshDesk | 372 | $1.30 | $1.44 | $2.96 | $4.60 | $10.30 |
| | Github | 1685 | $2.20 | $4.03 | $6.43 | $33.42 | $46.08 |
| | Onenote | 285 | $1.45 | $3.74 | $2.88 | $6.69 | $14.76 |
| | Optimizely | 208 | $1.18 | $3.81 | $2.84 | $6.73 | $14.56 |
| **Media** | Coursera | 751 | $1.51 | $1.52 | $1.29 | $13.83 | $18.15 |
| | Headspace | 1217 | $1.59 | $1.66 | $2.32 | $25.33 | $30.90 |
| | Health | 315 | $3.30 | $1.69 | $0.72 | $5.42 | $11.13 |
| | Spotify | 210 | $2.62 | $1.65 | $0.33 | $2.06 | $6.66 |
| | Youtube | 357 | $1.59 | $1.72 | $0.39 | $5.12 | $8.82 |
| **Communication** | Facebook | 36 | $1.99 | $1.55 | $2.70 | $0.43 | $6.67 |
| | Medium | 51 | $1.72 | $3.28 | $1.06 | $0.98 | $7.04 |
| | Microsoft Team | 35 | $1.77 | $1.48 | $2.09 | $0.52 | $5.86 |
| | Outlook | 186 | $1.37 | $3.39 | $1.10 | $2.78 | $8.64 |
| | Quora | 254 | $1.51 | $1.29 | $2.08 | $2.68 | $7.56 |
| | Signal | 226 | $2.08 | $1.74 | $2.01 | $2.90 | $8.73 |
| | Slack | 900 | $1.08 | $1.44 | $1.90 | $21.01 | $25.43 |
| | Zoom | 63 | $1.29 | $1.53 | $0.74 | $1.95 | $5.51 |
| **Total** | 29 | 11663 | $52.26 | $57.10 | $65.84 | $272.17 | $447.37 |

image at 12,845,056 (`--max_pixels 12845056`).

Training is run for 1 epoch (`--num_train_epochs 1`). We log every step (`--logging_steps 1`) and report metrics to Weights & Biases (`--report_to wandb`). We use DeepSpeed ZeRO-3 for memory-efficient sharding (`--deepspeed`, ZeRO-3 config). Checkpoints are saved with a step-based strategy (`--save_strategy steps`) and `--save_steps 1`, while storing only model weights (`--save_only_model true`). For GRPO sampling, we generate 8 rollouts per prompt (`--num_generations 8`).

### D.4. Reward Functions for GRPO

We use a composite reward consisting of three components: an action-type accuracy reward, a coordinate grounding reward, and a format reward. For a sampled rollout $y$ (the model completion) and its ground-truth solution $y^\star$, the overall reward is the sum of these components:

$$R(y, y^\star) = R_{\text{act}}(y, y^\star) + R_{\text{coord}}(y, y^\star) + R_{\text{fmt}}(y). \tag{6}$$

**Action-type accuracy reward.** Let $\text{act}(\cdot)$ be a parser that extracts the predicted action type from the `<action>...</action>` field (e.g., click, hover, drag, type_text, press_enter, scroll, answer,

*Table 9.* Unified GUI action space. Each action is represented as a JSON dictionary.

| **Action Format** |
|---|

```
{'action':  'click', 'coordinate':  [x, y]}
{'action':  'hover', 'coordinate':  [x, y]}
{'action':  'drag', 'from':  [x1, y1], 'to':  [x2, y2]}
{'action':  'type_text', 'text':  'content'}
{'action':  'press_enter'}
{'action':  'scroll', 'value':  down/up}
{'action':  'hotkey', 'value':  'Control + A}
{'action':  'hotkey', 'value':  'Delete'}
{'action':  'wait'}
{'action':  'answer', 'text':  'content'}
```

`hotkey`, `wait`). The action-type reward is a binary indicator of exact match:

$$R_{\text{act}}(y, y^\star) \;=\; \mathbb{I}[\text{act}(y) = \text{act}(y^\star)]. \tag{7}$$

**Coordinate grounding reward.**   For pointer-based actions (`click` and `hover`), we additionally verify whether the predicted coordinate falls inside the ground-truth bounding box. Let $\text{coord}(y) = (\hat{x}, \hat{y})$ be the extracted 2D coordinate from the predicted action, and let $\text{bbox}(y^\star) = (x_1, y_1, x_2, y_2)$ be the ground-truth bounding box. Since the model predicts coordinates in the resized space, we map them back to the original pixel space using per-example scale factors $s = (s_x, s_y)$:

$$(\tilde{x}, \tilde{y}) \;=\; (\lfloor s_x \hat{x} \rfloor, \lfloor s_y \hat{y} \rfloor). \tag{8}$$

The coordinate reward is then defined as

$$R_{\text{coord}}(y, y^\star) \;=\; \mathbb{I}[\text{act}(y) = \text{act}(y^\star)] \cdot \begin{cases} \mathbb{I}[x_1 \le \tilde{x} \le x_2 \,\wedge\, y_1 \le \tilde{y} \le y_2], & \text{if act}(y) \in \{\texttt{click}, \texttt{hover}\}, \\ 1, & \text{otherwise.} \end{cases} \tag{9}$$

That is, if the action type matches and the action is `click`/`hover`, we require the predicted point to lie inside the ground-truth box; for other action types, $R_{\text{coord}}$ reduces to action-type correctness.

**Format reward.**   To encourage structured outputs, we require the completion to follow the prescribed tag format: `<think>...</think><action>...</action>`. Let $\mathcal{F}$ denote the set of strings that fully match this template (implemented via a full-string regular-expression match). The format reward is

$$R_{\text{fmt}}(y) \;=\; \mathbb{I}[y \in \mathcal{F}]. \tag{10}$$

In practice, $\mathcal{F}$ corresponds to completions that contain both `<think>` and `<action>` blocks in the correct order with properly closed tags.

## E. Case Studies

In this section, we show the trajectory examples collected in AutoWebWorld.

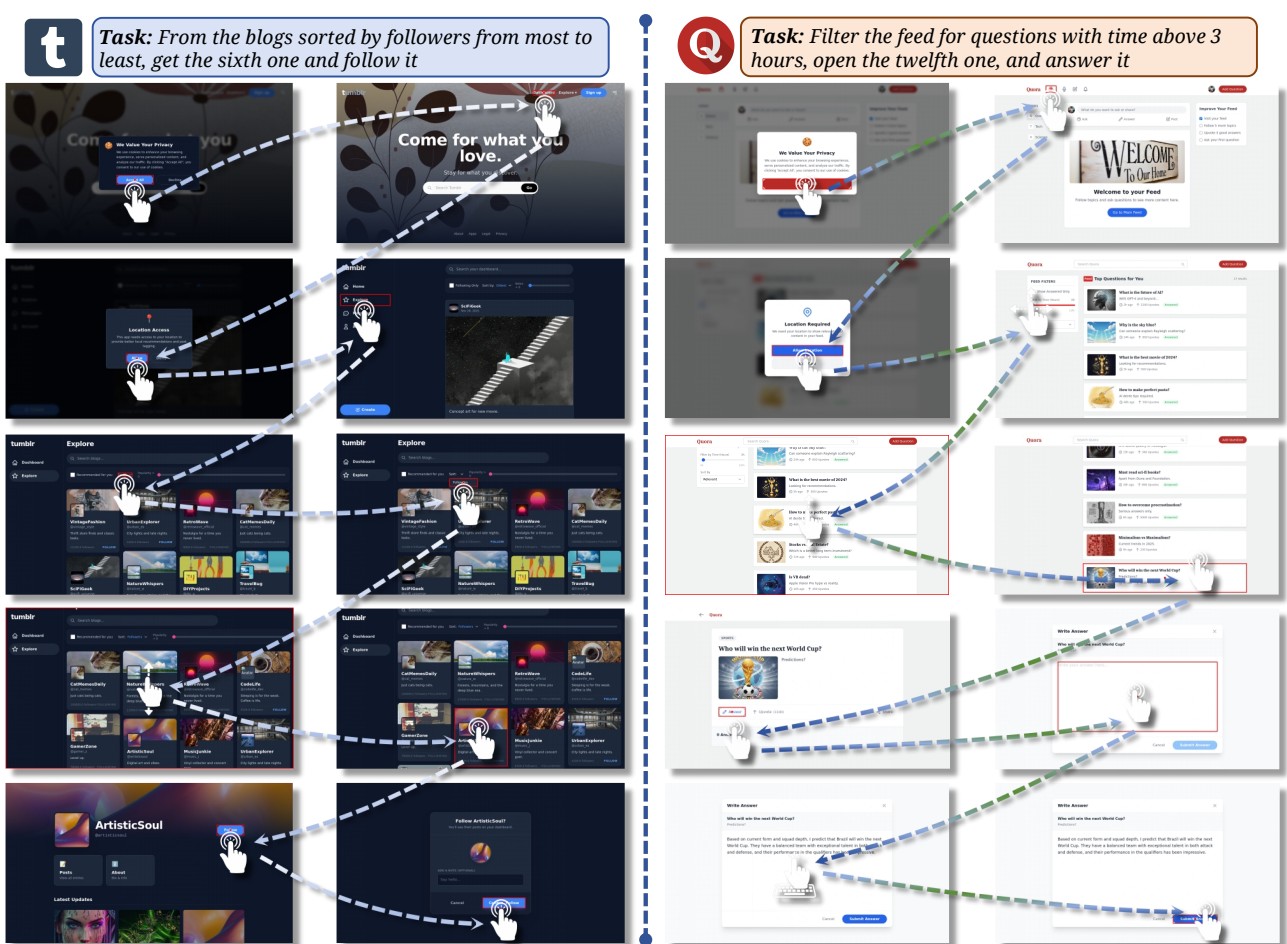

*Figure 6.* Trajectory example from cloned Tumblr and Quora websites.

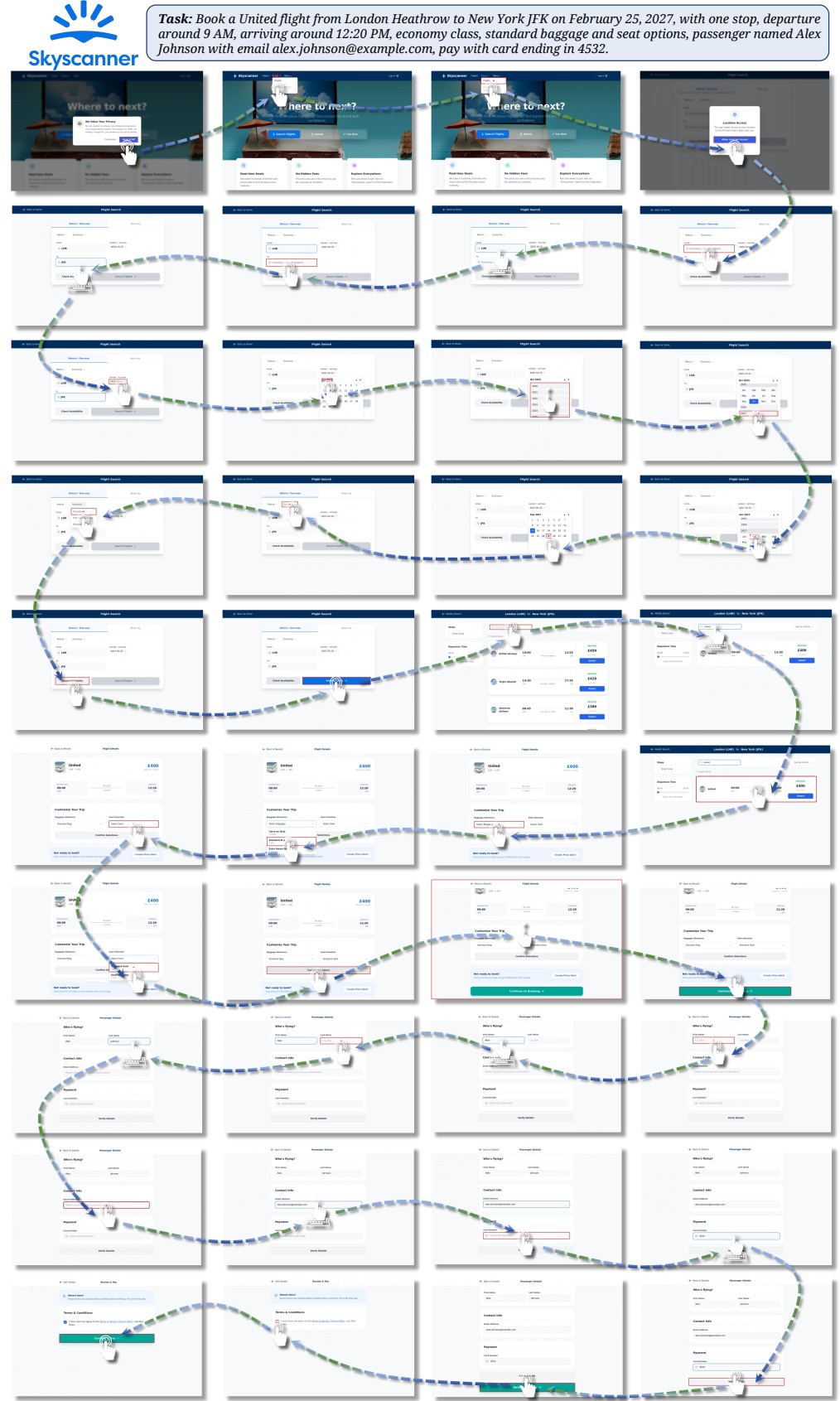

*Figure 7.* Trajectory example from cloned Skyscanner website.

