# OpenReview forum: "AutoWebWorld: Synthesizing Infinite Verifiable Web Environments via Finite State Machines"
_ICML.cc/2026/Conference — ICML 2026 regular_

### Official Review · Reviewer_Ckgk · 2026-03-05

**Soundness:** 2
**Presentation:** 3
**Significance:** 3
**Originality:** 4
**Overall Recommendation:** 4
**Confidence:** 4

**Summary:**

Interacting with real websites incurs high costs and is often difficult to verify. To address this issue, this paper proposes AutoWebWorld, a framework that is verifiable, scalable, and controllable. Based on this framework, approximately 29 web pages and 11,663 trajectories were synthesized. These trajectories were then used to train small models, and the resulting performance improvements demonstrate the effectiveness of the proposed framework.

**Compliance With Llm Reviewing Policy:**

Affirmed.

**Final Justification:**

Thank you to the authors for their response. My concerns have been addressed. Since I have already given a positive score of 4, I will maintain my current rating.

**Key Questions For Authors:**

(1) Can the generated websites simulate irreversible states? For instance, regarding a shopping cart, if an operation clears the cart, can the websites generated by the authors simulate this scenario?

(2) Some real websites are overly complex, where a single URL page might involve up to 100 interactive elements alongside complicated navigation between pages. If there are too many elements, what is your pruning principle? Will it inadvertently remove certain crucial navigation elements?

**Limitations:**

yes

**Strengths And Weaknesses:**

Strengths

(1)The cost analysis in this paper is highly satisfactory. Compared to previous methods, it significantly reduces construction costs.

(2)The finite state machines (FSMs) constructed in this paper are illuminating, stable, and controllable, which is crucial for agent training.

(3)Finally, the authors conducted a scaling analysis, proving the effectiveness of data volume in improving performance.

Weaknesses

(1) This paper lacks an analysis of the quality of the synthesized websites. When using Gemini 3 Pro to synthesize websites, do hallucinations and erroneous generations occur? Is there human verification regarding the proportion of such errors? Furthermore, if the navigation logic between pages is extremely complex, can the websites generated by large models fully simulate such complexity? Could the authors provide specific quantitative metrics to better demonstrate the generation quality of the simulated websites and make the paper more solid?

(2) Multiple stages of the current method rely on several commercial large models, such as GPT 5.1, Gemini 3 Pro, and DeepSeek V3.2. It remains unclear whether this framework would still be effective if advanced open source large models were used, which is critical for further expansion at a low cost.

(3) The current framework is only validated on a single model family. Its effectiveness across different model families remains unclear, such as the LLaVA series.

---

> ### Author Rebuttal · Authors · 2026-03-31
>
> We thank the reviewer for recognizing the originality of our work and the constructive feedback.
>
> ## W1: Lack of quality analysis for synthesized websites
>
> We provide quantitative evidence from two angles:
>
> **Trajectory execution success rate.** After BFS generates candidate trajectories, we replay each one on the synthesized website using Playwright (Section 4.4). A trajectory passes only if every step executes successfully and the final state matches the intended goal. Across our environments, the overall pass rate is **85.3%**. 8 environments achieve 100% pass rate (JD_COM, Shopee, Walmart, Facebook, Zoom, Headspace, Airtable, Skyscanner). This metric directly measures functional correctness — if a website has hallucinated elements or broken navigation, the corresponding trajectories would fail during replay. Failed trajectories are automatically discarded by our filtering pipeline, so only verified ones enter training.
>
> For the 4 environments below 60% (Microsoft Teams 15.7%, Asana 20.3%, Medium 30.9%, Freshdesk 53.8%), failures are mainly caused by semantic mismatches where the coding agent did not fully implement certain FSM-specified interactions. Our self-repair loop (Stage 4) catches build-level errors but not all semantic mismatches. These environments still contribute verified trajectories from their successfully executed paths.
>
> **Visual similarity to real websites.** We compared homepage screenshots of synthesized websites against their real-world counterparts using CLIP similarity and Gemini-3-Flash scoring. Average CLIP score: **0.7295**, average VLM score: **0.81/1.0** across 13 websites (full table in supplementary). Notably, the coding agent only receives textual references — no reference screenshots.
>
> ## W2: Dependence on commercial large models
>
> Our pipeline is model-agnostic by design — each stage uses an LLM/coding agent through a standard API, with no architecture-specific coupling. GPT-5.1 (FSM generation), Gemini-3-Pro (web generation), and DeepSeek-V3.2 (query generation) can each be replaced by any sufficiently capable model. As open-source models continue to improve in coding and instruction-following, we expect them to become viable drop-in replacements. We chose commercial models for the initial implementation to maximize generation quality, but the framework itself does not depend on them.
>
> ## W3: Only validated on Qwen model family
>
> We note that in the current GUI agent landscape, Qwen2.5-VL has become the de facto base model — all baselines in our paper (UI-TARS, OpenCUA, TongUI, UI-Venus) are built on Qwen. This reflects a community consensus rather than a limitation specific to our work. That said, we agree exploring other base models is valuable and plan to validate on additional families (e.g., InternVL) in future work. Within Qwen, we already show consistent improvements across two scales (3B and 7B), suggesting gains are not configuration-specific.
>
> ## Key Questions
>
> **Q1: Can generated websites simulate irreversible states?** Yes. In our FSM, each state is defined by (page_id, signature). When an irreversible action occurs — e.g., clearing a shopping cart — the corresponding signature variable (e.g., cart_items) is updated to null via the action's effect rule, and no other action's effect can restore it. This is enforced at the FSM level: if no action defines an effect that repopulates the cart, the state transition is inherently irreversible. Similarly, terminal states like ORDER_SUBMITTED_SUCCESS have no outgoing transitions by design.
>
> **Q2: Pruning principle for complex pages with many elements?** We designed three complexity profiles (easy, medium, hard) in the FSM specification, each controlling the number of pages and actions per page. Through experimentation, we found the medium profile (10-20 interactive elements per page) produces the most stable generation results. This is not random pruning — the FSM proposer selects task-critical elements (navigation, forms, filters, buttons) based on the reference website's core workflows, while the complexity profile bounds the total count. The FSM validator further checks that all terminal states remain reachable, ensuring no crucial navigation paths are removed. For real websites with 100+ elements, our approach focuses on the functional subset that matters for task completion, which aligns with how users actually interact with complex pages.

---

> > ### Author Rebuttal · Reviewer_Ckgk · 2026-04-04
> >
> > Thank you to the authors for their response. My concerns have been addressed. Since I have already given a positive score of 4, I will maintain my current rating.

---

> > > ### Author Response · Authors · 2026-04-05
> > >
> > > We thank you for confirming that the concerns have been fully addressed. We appreciate your recognition of the originality of our work and the constructive questions you raised.

---

### Official Review · Reviewer_eKum · 2026-03-10

**Soundness:** 4
**Presentation:** 2
**Significance:** 3
**Originality:** 3
**Overall Recommendation:** 4
**Confidence:** 4

**Summary:**

This paper tackles the cost and unreliability of collecting verifiable interaction trajectories for Web GUI agent training. Real websites hide their internal state, so verifying whether an action sequence is correct requires human annotators or LLM judges. AutoWebWorld sidesteps this by constructing synthetic web environments whose internal state is fully specified as a Finite State Machine (FSM). Because states, actions, and transitions are explicit, correctness reduces to checking whether an action sequence reaches a goal state in the FSM graph.

The pipeline has four stages: a multi-agent LLM system generates an FSM specification from a website theme; a coding agent translates it into a working web app; BFS over the state graph enumerates candidate trajectories; and candidates are replayed via Playwright and filtered for execution correctness. The authors build 29 environments, collect 11,663 verified trajectories, and train Qwen2.5-VL (3B/7B) via GRPO. They report competitive WebVoyager results with substantially less training data than prior work, along with a monotonic scaling trend.

**Compliance With Llm Reviewing Policy:**

Affirmed.

**Ethical Review Concerns:**

This paper addresses the challenge of collecting high-quality, verifiable interaction trajectories for training autonomous Web GUI agents. The core observation is that real websites are "black boxes" whose internal state is hidden from the agent, making it expensive and unreliable to verify whether a given action sequence is correct — typically requiring human annotators or LLM judges. The proposed solution, AutoWebWorld, flips this by constructing synthetic web environments where the internal state is fully specified as a Finite State Machine. Because all states, actions, and transitions are explicit, correctness can be checked programmatically by testing whether an action sequence reaches a designated goal state in the FSM graph.

The pipeline works in four stages: a multi-agent LLM system proposes and validates an FSM specification for a given website theme; a coding agent translates that FSM into a working web application; BFS over the state graph enumerates candidate trajectories; and those candidates are replayed on the generated website and filtered for execution correctness. The authors build 29 environments, collect 11,663 verified trajectories, and use them to train Qwen2.5-VL models (3B and 7B) via GRPO. They evaluate primarily on WebVoyager and report competitive results with substantially less training data than prior work, along with a monotonic scaling trend relating synthetic data volume to downstream performance.

**Final Justification:**

keeping my score, see rebuttal ack

**Key Questions For Authors:**

Can you provide multi-seed results (at least 3 seeds) for the main WebVoyager comparison and the scaling curve?
How exactly are GRPO rollouts generated from offline trajectory data? Is this step-level offline RL, or does the model interact with environments at training time?
What is the distribution of goal-state types across the 29 environments?
What are actual state-space sizes for representative environments, and what fraction does BFS explore?
Have you tried an SFT baseline on the same data to disentangle the data contribution from the training algorithm?

**Limitations:**

See Weaknesses

**Strengths And Weaknesses:**

## Strengths

Novel and well-motivated formulation. Modeling web environments as FSMs to make verification intrinsic rather than outsourced to external judges is elegant and well-argued. The formalization in Section 3 is clean, and the distinction between agent observations and internal semantic state is a useful conceptual contribution.
End-to-end automation. The full pipeline requires no human annotation, a meaningful practical advance over approaches relying on human demonstrators or LLM judges.
Data efficiency. Achieving competitive WebVoyager performance with 16K GRPO steps, compared to systems trained on 350K to 1M samples, is a striking result. The scaling curve in Figure 4 suggests room to grow.
Longer-horizon trajectories. The average trajectory length of ~22 steps substantially exceeds prior datasets (7-12 steps), improving coverage of compositional, multi-step tasks.
Dual use as benchmark. Table 6 shows frontier models score lower on AutoWebWorld than WebVoyager, validating that the synthesized environments are non-trivial.

## Weaknesses

### Major

W1: No uncertainty estimates. This is the most critical concern. No results include standard errors, confidence intervals, or variance estimates. WebVoyager evaluation involves few tasks per domain (some under 50), making results sensitive to random variation. The GRPO curve in Figure 5 appears to be from a single seed. Without multi-seed runs, it is unclear whether the reported improvements are robust. The authors should report results over 3-5 seeds with standard errors for Tables 3-5 and Figure 4.

W2: Task and goal-state generation is underspecified. The FSM proposer (GPT-5.1) decides what constitutes a goal state and thus what tasks each environment supports. This is arguably the most consequential design decision in the pipeline, since a training environment is only as good as its tasks, yet it receives minimal discussion. How diverse are the resulting goals? Are they biased toward simple terminal conditions ("reached confirmation page") while missing subtler completions? An analysis of goal-state diversity, or at least a taxonomy of generated task types, would strengthen the paper.

W3: Combinatorial state-space explosion is insufficiently addressed. The paper mentions BFS deduplication and depth caps (Appendix A.2.2) but provides no analysis of how state spaces grow in practice. With multiple pages and multi-valued signature variables, reachable states grow combinatorially. The authors should report actual state-space sizes for their 29 environments, discuss how caps were chosen, and analyze what fraction of the theoretical state space BFS covers.

W4: Connection between offline trajectories and GRPO is unclear. GRPO learns from sampled rollouts and their rewards, but the training data consists of pre-collected offline trajectories. The paper does not explain how these are reconciled. Are trajectories used as step-level prompts from which the model samples 8 completions? Or does the model roll out in the environments online? This distinction matters for interpreting what the model learns. Section 5.1.1 and Appendix D should clarify this.

### Minor

W5: Cost comparison is misleading. Table 1 compares per-trajectory costs across methods from different time periods. LLM inference costs have dropped dramatically, making cross-era dollar comparisons uninformative. The $0.04 figure also amortizes $447 of pipeline cost over all 11,663 trajectories, but only 1,215 are actually used for training, yielding $0.37 per used trajectory.

W6: Promotional language. Phrases like "Remarkably, our agents achieved state-of-the-art" (line 37) read as promotional. The results can speak for themselves.

W7: FSM expressiveness limitations. The formulation assumes deterministic transitions and finite variable domains. Real websites involve stochastic elements (recommendations, A/B testing), continuous inputs (free-form text), and context-dependent behavior. A discussion of what the FSM formalism cannot capture, and how this limits transfer, would be valuable.

W8: Evaluation scope. WebVoyager is evaluated on only 9 websites after filtering for stability. The paper should report what fraction of the original benchmark was excluded and whether excluded sites differ systematically.

I think this paper should be a 5. Going to give a score of 4 while till the issues are adressed.

---

> ### Author Rebuttal · Authors · 2026-03-31
>
> We thank the reviewer for the detailed review and are encouraged that you consider this paper worthy of a score of 5.
> ## Major
> **W1:** We ran 3 seeds for all main experiments:
>
> *WebVoyager (%):*
>
> | | Apple | Arxiv | Coursera | CD | BBC | GF | GM | HF | WA | Overall |
> |---|---|---|---|---|---|---|---|---|---|---|
> | Run 1 | 25.00 | 21.43 | 30.00 | 60.47 | 31.25 | 7.32 | 15.00 | 32.43 | 23.91 | 27.42 |
> | Run 2 | 31.25 | 21.43 | 25.00 | 72.09 | 21.88 | 0.0 | 10.00 | 32.43 | 30.43 | 27.17 |
> | Run 3 | 34.38 | 16.67 | 27.50 | 74.42 | 37.50 | 0.0 | 12.50 | 21.62 | 26.09 | 27.85 |
> | Mean±Std | 30.21±4.78 | 19.84±2.75 | 27.50±2.50 | 68.99±7.47 | 30.21±7.86 | 2.44±4.23 | 12.50±2.50 | 28.83±6.24 | 26.81±3.32 | **27.48±0.35** |
>
> *ScreenSpot-V2:*
>
> | | Mobile Text | Mobile Icon | Desktop Text | Desktop Icon | Web Text | Web Icon | Overall |
> |---|---|---|---|---|---|---|---|
> | Ours-3B | 70.11±1.60 | 53.40±0.97 | 76.80±2.76 | 40.00±1.17 | 67.66±3.05 | 64.37±1.52 | 63.68±1.58 |
> | Ours-7B | 95.86±0.49 | 78.67±0.39 | 89.52±1.06 | 70.00±1.54 | 91.03±0.92 | 78.49±0.61 | 85.53±0.46 |
>
> *ScreenSpot-Pro:*
>
> | | Icon | Text | Average |
> |---|---|---|---|
> | Ours-3B | 4.36±0.55 | 26.95±2.82 | 18.32±1.89 |
> | Ours-7B | 8.67±0.16 | 44.78±1.84 | 29.54±1.77 |
>
> GRPO training curves across 3 seeds are shown in https://ibb.co/F4XXF35m.
>
> **W2: Goal-state generation underspecified.** GPT-5.1 generates goals conditioned on both web theme and reference website name. We evaluated all 145 terminal goals using three independent LLMs, each scoring on diversity (coverage of user intents) and match (alignment with the reference website's typical tasks). Full results are in  https://ibb.co/VYwKJtXb.
>
> **W3: State-space explosion.** Full per-environment BFS statistics are in https://ibb.co/XZSLjwcT. We clarify the metrics:
>
> *State space:* For each page, we compute the Cartesian product of all signature variable domains (determined by action effects and data.js entries). Open-ended text inputs are collapsed to a single placeholder value, contributing a factor of 2 (null vs. filled). The total environment state space is the sum across all pages.
>
> *BFS visited nodes:* The number of unique (page_id, signature_hash) pairs that BFS dequeued and expanded — i.e., distinct semantic states actually processed.
>
> *Key findings:* BFS achieves ≥95% coverage in 20 out of 27 environments, with many reaching 100%. For environments with lower coverage, there are two main reasons. First, our BFS selects the shortest path to each terminal state, so optional in-page actions not on any shortest path (e.g., starring a comment, toggling a non-essential filter) are naturally excluded. Second, some generated pages may not be fully connected to the navigation graph, making portions of their state space unreachable. Both cases do not affect trajectory quality: all defined terminal goals remain reachable.
>
> **W4: Offline trajectories and GRPO.** We use step-level GRPO (following WebFactory). BFS trajectories are decomposed into single-step samples (prompt = screenshot + history; ground truth = action type + bounding box for reward). Model samples 8 completions per prompt; trajectories serve as prompt sources and reward references.
> ## Minor
>
>  **W5: Cost comparison.** Acknowledged. $0.04 = total cost / all trajectories; per used cost is 0.37 (1,215 used). The reason we used only a subset is GRPO training efficiency, training on even the sampled data takes over 30 hours on 8×A800 GPUs. We plan to train on the full trajectory set in future work and expect further performance gains based on the scaling trend in Figure 4.
>
> **W6: Promotional language.** Agreed. Will revise phrasing like "Remarkably" to let results speak for themselves.
>
> **W7:** Determinism enables intrinsic verification — a deliberate trade-off. Our SOTA on real dynamic websites confirms effective transfer despite this.
>
> **W8:** 5 websites excluded (CAPTCHA/access-denied); Benchmarks like MiniWob++ also show that our models are stronger than baselines and learn the general GUI abilities.
>
> ## Key Questions
>
> SFT baseline: **SFT 5.34% vs GRPO 27.42%**. SFT's low efficiency stems from two factors: (1) SFT trains on the entire output including lengthy thinking tokens, but for GUI agents the key signal is the ground-truth coordinate — thinking tokens dilute learning of the core grounding skill, while GRPO rewards focus directly on action type and coordinate accuracy; (2) SFT targets the bbox center point, but any point within the bbox is a valid click — this introduces systematic bias, whereas GRPO accepts any prediction within the bbox. Other questions addressed in W1-W4.

---

> > ### Author Rebuttal · Reviewer_eKum · 2026-04-03
> >
> > We thank the authors for the responsive rebuttal.
> >
> > **Well addressed.** The 3-seed results resolve our most critical concern. Overall WebVoyager variance is tight (27.48±0.35), though per-domain variance remains notable in some cases. The step-level GRPO clarification (W4) fills an important gap and should be in the main text. The SFT vs GRPO comparison (5.34% vs 27.42%) and the coordinate bias explanation are insightful contributions that deserve space in the paper.
> >
> > **Partially addressed.** The goal-state diversity evaluation (W2) uses LLMs to validate LLM-generated outputs, which is circular. A simple human spot-check or goal-type taxonomy with counts would be more convincing. The state-space analysis (W3) is helpful, but collapsing open-ended text inputs to binary (null vs. filled) is a meaningful simplification that should be flagged as a limitation. On FSM expressiveness (W7), citing WebVoyager performance as evidence of transfer is circular when WebVoyager is the evaluation target. A brief discussion of what real-world web behavior the FSM formalism systematically misses would be more useful.
> >
> > **Still open.** The evaluation scope response (W8) does not characterize whether the 5 excluded websites differ systematically from retained ones.
> >
> > I'm maintaining our positive score. The paper presents a well-motivated formulation with strong empirical backing, and the rebuttal has addressed the key reproducibility concerns.

---

> > > ### Author Response · Authors · 2026-04-07
> > >
> > > We thank you for the thoughtful follow-up and are glad that the key reproducibility concerns (W1, W4, SFT baseline) are well addressed. We respond to the remaining points below.
> > >
> > > **W2 (circular evaluation).** We agree that LLM-evaluating-LLM is not ideal. To address this without relying on any LLM judgment, we manually constructed a two-level goal taxonomy with 6 coarse-grained categories and 14 fine-grained subcategories, then had team members classify all 145 terminal goals into this taxonomy and verify each goal against its reference website for realism:
> > >
> > > | Category | Subcategory | Count | % |
> > > |---|---|---|---|
> > > | **Transaction** | Checkout / Order Placement | 19 | 13.1% |
> > > | | Payment / Financial Transfer | 7 | 4.8% |
> > > | | Subscription / Enrollment | 8 | 5.5% |
> > > | **Content Creation** | Record / Item Creation | 24 | 16.6% |
> > > | | Content Publishing / Posting | 6 | 4.1% |
> > > | | Review / Rating / Feedback | 12 | 8.3% |
> > > | **Communication** | Email / Direct Messaging | 14 | 9.7% |
> > > | | Meeting / Call | 7 | 4.8% |
> > > | **Account Mgmt** | Profile / Account Settings | 14 | 9.7% |
> > > | **Booking** | Booking / Reservation | 14 | 9.7% |
> > > | | Search / Alert Setup | 4 | 2.8% |
> > > | **Operations** | Task / Workflow Management | 12 | 8.3% |
> > > | | Code / Dev Operations | 4 | 2.8% |
> > > | | Browse / Consume Content | 5 | 3.4% |
> > >
> > > 6 coarse categories, 14 fine-grained subcategories, no single subcategory exceeds 17%. The uniform distribution across diverse functional types — spanning transactions, content creation, communication, account management, booking, and operations — confirms that our goal generation is not biased toward any single pattern. All classification and verification were performed by human team members without LLM involvement.
> > >
> > > **W3 (binary simplification).** Thank you for pointing this out — we agree that collapsing open-ended text inputs to binary (null vs. filled) is a meaningful simplification and should be transparently acknowledged. We will explicitly flag this as a limitation in the revision. That said, we want to clarify that this simplification only affects the BFS coverage statistics reported in our state-space analysis; the actual training data retains diverse text content, as placeholders are instantiated with varied realistic text during the query generation stage.
> > >
> > > **W7 (FSM expressiveness).** We provide the requested discussion of what the FSM formalism systematically misses: (1) stochastic content — recommendation algorithms and A/B testing cause non-deterministic responses to identical actions; (2) real-time updates — push notifications, price fluctuations, and inventory changes; (3) free-form text semantics — search ranking depends on query meaning, not just string matching; (4) session dynamics — login timeouts, cookie expiration, rate limiting; (5) cross-site interactions — OAuth redirects, payment gateway callbacks. These are inherent trade-offs of the deterministic FSM design that enables intrinsic verification. We will add this discussion to the revision.
> > >
> > > **W8 (excluded websites).** The 5 excluded websites were removed purely due to technical access issues. We provide two analyses showing no systematic bias:
> > >
> > > *No difficulty gap.* GPT-4 success rates from `the original WebVoyager paper` show both sets contain easy and hard sites. Excluded: Allrecipes 11.1%, Amazon 17.1%, Booking 22.7%, GitHub 48.8%, Google Search 60.5%. Retained includes some of the hardest sites in the benchmark: Google Flights 2.4%, BBC 9.5%, Arxiv 14.0%.
> > >
> > > *No coverage gap.* Each excluded website's functional type is represented by a retained website: Amazon (e-commerce) → Apple; GitHub (code hosting) → HuggingFace; Google Search (search/query) → Wolfram Alpha, Arxiv; Booking (travel/reservation) → Google Flights, Google Maps; Allrecipes (content browsing) → Coursera, BBC. No interaction category is missing from our evaluation set.
> > >
> > > We appreciate this suggestion and will explore alternative approaches (e.g., proxy services, headless browser configurations) to recover results for the excluded websites in future work. Additionally, our MiniWoB++ evaluation (0%→80% for 3B, 10%→90% for 7B, reported in our response to another reviewer) provides further evidence of generalization on an entirely independent GUI benchmark that does not suffer from access issues.

---

### Official Review · Reviewer_GgDV · 2026-03-12

**Soundness:** 3
**Presentation:** 3
**Significance:** 3
**Originality:** 3
**Overall Recommendation:** 3
**Confidence:** 4

**Summary:**

This paper proposes AutoWebWorld, a framework for generating synthetic, verifiable web environments by first specifying websites as finite state machines (FSMs), then using coding agents to translate these FSMs into executable websites, and finally enumerating and filtering valid trajectories through BFS plus Playwright replay. The central claim is that explicit state-transition structure removes the “verifier bottleneck” in web-agent training data collection, enabling cheaper, more scalable, and more reproducible supervision than collecting trajectories from real websites with humans or LLM judges. Empirically, the authors build 29 environments, generate 11,663 verified trajectories at a reported cost of $0.04 per trajectory, and show improved results on WebVoyager, ScreenSpot-V2, and ScreenSpot-Pro, along with a synthetic-data scaling trend on WebVoyager and Online-Mind2Web.

**Compliance With Llm Reviewing Policy:**

Affirmed.

**Final Justification:**

The clarification and new results fully address my concerns, so I would improve the score to 4.

**Key Questions For Authors:**

1. Since WebVoyager evaluation uses only 9 websites and an LLM judge, can you clarify how much this affects comparison to prior reported numbers?

**Limitations:**

The authors are suggested to add experiments or discussions on the Synthetic-to-real gap. For instance, finite-state abstractions may underspecify richer real web behaviors such as unexpected side effects, long-tail interaction patterns, or open-ended user goals.

**Strengths And Weaknesses:**

Strengths:
1. The paper addresses a real and important bottleneck in web-agent research: step-level verification on real sites is expensive, noisy, and hard to scale. The FSM formulation is conceptually clean, and the full pipeline from FSM generation to website synthesis to BFS search to execution-based filtering is easy to follow. The distinction between internal semantic state and rendered observation is also well motivated.

2. The proposed method works effectively. The reported gains on WebVoyager are substantial for a 7B model trained with only about 16k GRPO steps, and the grounding improvements on ScreenSpot-V2 and ScreenSpot-Pro suggest that the synthetic environments are not only useful for navigation but also for UI grounding. The paper also does more than a single-point comparison: it includes a scaling study, an ablation on the importance of grounding data, and a small benchmark-style comparison showing that AutoWebWorld tasks are not obviously trivial.

3. Compared with prior datasets that rely on external verification, AutoWebWorld’s “inherent verifier” framing is a meaningful systems contribution, even if some parts of the pipeline still use strong external models for generation. Breaking down the generation cost by stage is helpful.

Weaknesses:
1. Although the paper argues that synthetic verified data transfers to real web navigation, the environments are still generated from author-designed FSMs and then implemented by coding agents. This creates a large abstraction gap between AutoWebWorld and the partial observability, brittleness, latency, and unexpected side effects of real websites. The paper shows transfer to WebVoyager, but it is still unclear how much of the gain comes from learning generally useful web skills versus learning biases of the synthetic generation pipeline.

2. The task success on WebVoyager and Online-Mind2Web is determined by Gemini-3-Flash from the instruction and interaction trace, rather than an exact environment-level success signal. While this is common in web-agent evaluation, it introduces potential judge noise and weakens the paper’s broader claim of eliminating verifier ambiguity, since the real-world transfer results still rely on an external LLM judge. Also, it is unclear whether the Gemini-3-Flash judgment is accurate. Adding human verification or evaluating on datasets with a programmatic verifier (e.g., OSWorld's subsets) might strengthen this.

3. WebVoyager evaluation is restricted to 9 “stable” websites, with the rest filtered out due to access-denied and CAPTCHA issues. That is understandable operationally, but it also makes the benchmark setting narrower than the headline claim might suggest.

---

> ### Author Rebuttal · Authors · 2026-03-31
>
> We thank the reviewer for the thorough and constructive review.
>
> ## W1: Abstraction gap between synthetic and real environments
>
> The reviewer raises a key question: do gains come from learning general web skills or from biases of the synthetic pipeline? We provide two pieces of evidence:
>
> **1. MiniWoB++ evaluation.** To directly test whether our models acquire general GUI skills, we evaluated on MiniWoB++, a standard benchmark for fundamental web interaction abilities (clicking, typing, scrolling, form-filling, etc.) that is entirely independent from our training environments:
>
> | | Buy-Ticket | Click-btn | Checkbox | Click-link | Scroll-list | Enter-text-2 | Enter-text | Login | Scroll | Search | SR |
> |---|---|---|---|---|---|---|---|---|---|---|---|
> | Qwen-3B | 0 | 0 | -1 | 0 | 1 | 0 | -1 | -1 | -1 | 0 | 0% |
> | Ours-3B | 1 | 1 | 1 | 1 | 1 | 1 | 1 | 1 | -1 | -1 | 80% |
> | Qwen-7B | 0 | 1 | 0 | 0 | 0 | 0 | 0 | 0 | -1 | 0 | 10% |
> | Ours-7B | 1 | 1 | 1 | 1 | 1 | 1 | 1 | 1 | 1 | -1 | 90% |
>
> The 0%→80% (3B) and 10%→90% (7B) improvements on a completely independent benchmark confirm that AutoWebWorld teaches broadly transferable GUI skills, not synthetic-environment-specific patterns.
>
> **2. Cross-domain consistency.** In Table 3, our model improves over the baseline across all 9 diverse WebVoyager websites spanning different domains (e-commerce, news, academic, travel, etc.). If gains were due to pipeline biases, we would expect improvements only on domains similar to our training environments. The consistent cross-domain gains indicate genuine skill transfer.
>
> ## W2: Gemini-3-Flash judge undermines the "no external verifier" claim
>
> We appreciate this observation and clarify the scope of our claim. Our contribution of *inherent verification* applies to the **training data collection** stage: unlike prior pipelines that require external judges to verify each trajectory, AutoWebWorld generates trajectories that are correct by construction via FSM state transitions. This eliminates the verifier bottleneck during data synthesis.
>
> For **evaluation**, we follow the community standard protocol for WebVoyager—all prior works (UI-TARS, OpenCUA, TongUI, etc.) use LLM judges for this benchmark since it operates on live websites without programmatic success criteria. This is orthogonal to our training-time contribution.
>
> To further address this concern, we additionally evaluated on **OSWorld's Chrome tasks** (46 web-related tasks), which uses a programmatic verifier with no LLM judge involved:
>
> | Model | OSWorld Chrome (%) |
> |---|---|
> | Qwen2.5-VL-7B | 2.17 |
> | Ours-7B | 17.39 |
>
> The +15.22% improvement under programmatic verification confirms our gains are real and not artifacts of LLM judge noise. Moreover, two additional evaluations require **no external judge at all**:
>
> `(1)`Table 6 evaluates frontier agents on AutoWebWorld environments using our intrinsic FSM-based verifier, demonstrating that our environments can serve as judge-free benchmarks;
>
> `(2)` the MiniWoB++ evaluation in W1 also uses programmatic success criteria, and our model shows 10%→90% (7B) improvement without any LLM judge involvement.
>
> ## W3: WebVoyager limited to 9 websites
>
> We excluded 5 websites from the original WebVoyager benchmark: Allrecipes, Amazon, GitHub, Booking, and Google Search. All were excluded due to persistent access-denied errors or CAPTCHA redirects during evaluation, making reliable assessment infeasible. We apply the **same 9-website subset to all models** in Table 3, so comparisons remain apple-to-apple. We will explicitly report the excluded websites and reasons in the revision.
>
> ## Key Question: Comparability with prior reported numbers
>
> All baselines in Table 3 are evaluated on the same 9 websites with the same judge, so comparisons are internally consistent. Regarding comparability with prior work: we use a strict 15-step budget, while recent works such as Fara-7B cap at 100 steps for all online benchmarks. To show our model's capability extends beyond 15 steps, we ran an additional 30-step evaluation:
>
> | | Apple | Arxiv | Coursera | CD | BBC | GF | GM | HF | WA | Overall |
> |---|---|---|---|---|---|---|---|---|---|---|
> | Ours 15s | 25.00 | 21.43 | 30.00 | 60.47 | 31.25 | 7.32 | 15.00 | 32.43 | 23.91 | 27.42 |
> | Ours 30s | 34.38 | 38.09 | 32.50 | 74.42 | 56.25 | 0.0 | 25.00 | 51.35 | 41.30 | 39.25 |
>
> With 30 steps, performance improves substantially (+11.83%), showing that our model's capability is underestimated at 15 steps. Per-website results become directly comparable to prior work that uses larger step budgets.

---

> > ### Author Rebuttal · Reviewer_GgDV · 2026-04-04
> >
> > The clarification and new results fully address my concerns.

---

> > > ### Author Response · Authors · 2026-04-05
> > >
> > > We sincerely thank the reviewer for confirming that the concerns have been fully resolved. We are glad that the clarifications and new experiments addressed all the raised issues. Given that the reviewer selected "(a) Fully resolved - My concerns have been adequately addressed" and noted that "the clarification and new results fully address my concerns," we kindly ask if the reviewer would consider reflecting this in the score. We greatly appreciate the reviewer's time and constructive feedback throughout this process.

---

### Official Review · Reviewer_5Uut · 2026-03-13

**Soundness:** 3
**Presentation:** 3
**Significance:** 3
**Originality:** 3
**Overall Recommendation:** 4
**Confidence:** 3

**Summary:**

This paper proposes AutoWebWorld, a framework for generating verifiable training trajectories at scale by modeling web environments as finite state machines. The system first generates an FSM specification of a website, then synthesizes a corresponding web environment and finally enumerates valid interaction trajectories through graph search. Since by design this is fully observable, the resulting trajectories can be verified by construction without external judging. Experiments show training on these synthetic trajectories improve performance on several benchmarks.

**Compliance With Llm Reviewing Policy:**

Affirmed.

**Final Justification:**

I recommend a weak accept. Overall, I value that this paper tackles an important problem in automated training data generation: how to cheaply generate verifiable web interaction trajectories *at scale*. Although the simplified nature of the websites remains a concern, the new experiments in the rebuttal addressed my main concern regarding the paper’s central motivation. I therefore believe the strengths outweigh the weaknesses.

**Key Questions For Authors:**

How does training on AutoWebWorld data compare with training on existing trajectory datasets such as Explorer?

How well do models trained on AutoWebWorld data transfer to real-world websites that involve dynamic content and asynchronous updates?

How realistic are the generated FSM environments compared to real web interaction?

**Limitations:**

See review.

**Strengths And Weaknesses:**

Strengths:

This paper tackles an important problem in automated training data generation: how to cheaply generate verifiable web interaction trajectories. By characterizing website interactions as transitions in a FSM, the framework is able to generate at scale without external judges.


Weaknesses:
1. The main concern is that the experimental results do not strongly support the central motivation, i.e., the importance of verifiability. The main narrative of the paper is that real-world web trajectories are difficult to verify and therefore introduce noise into training data. However, the experiments do not clearly demonstrate that this issue is a major bottleneck in practice. In particular, the paper does not directly compare training with verified trajectories versus training with noisy trajectories collected from real websites. Without this comparison, it is unclear whether it is necessary to do this over existing datasets that already scale despite imperfect supervision.

2. The other concern is that characterizing websites as FSMs seems limiting. Real-world websites often involve dynamic content or asynchronous updates that may not be well captured by a simple FSM formulation. As a result, the generated environments may fail to capture important aspects of real web interaction.

---

> ### Author Rebuttal · Authors · 2026-03-31
>
> We sincerely thank the reviewer for the constructive feedback. We address each concern below.
>
> ## W1: Experimental results do not strongly support the importance of verifiability
>
> We agree this comparison was missing and have now conducted a direct head-to-head experiment.
>
> **Experiment setup.** Following the Explorer paper's data collection protocol, we used GPT-4o to crawl 1,500 popular websites and collected 1,500 trajectories ( around 10K steps) via agent self-exploration, at a cost of 180$ ( 0.12 per trajectory). For fair comparison, we also retrained our model using only ~10K steps of AutoWebWorld data. Both models were trained with identical GRPO pipelines on Qwen2.5-VL-7B-Instruct.
>
> **Results on WebVoyager (15 steps, %):**
>
> | | Apple | Arxiv | Coursera | CD | BBC | GF | GM | HF | WA | Overall |
> |---|---|---|---|---|---|---|---|---|---|---|
> | Qwen2.5-7B | 15.62 | 2.38 | 2.50 | 16.28 | 6.25 | 0.0 | 0.0 | 0.0 | 4.35 | 5.26 |
> | Explorer | 12.50 | 2.38 | 21.88 | 55.81 | 22.50 | 0.0 | 2.50 | 10.81 | 26.09 | 17.56 |
> | Ours | 21.88 | 16.67 | 30.00 | 72.09 | 37.50 | 0.0 | 7.50 | 24.32 | 23.91 | 25.99 |
>
> AutoWebWorld outperforms Explorer by **+8.43%** overall with matched training steps, while costing significantly less (0.04 vs 0.12$ per trajectory). The gains are consistent across domains, with particularly large improvements on Arxiv (+14.29%), BBC (+15.00%), and HuggingFace (+13.51%). This demonstrates that intrinsically verified synthetic trajectories provide stronger supervision than externally-verified real-world trajectories, directly supporting our core motivation.
>
> ## W2: Characterizing websites as FSMs seems limiting
>
> We acknowledge that FSMs cannot model all dynamic aspects of real websites. However, we highlight three points:
>
> **(1) Determinism is a deliberate design choice.** Intrinsic verification—our core contribution—requires that state transitions be deterministic and fully observable. This is the enabling condition for verification-by-construction, not merely a simplification.
>
> **(2) We already handle common edge cases.** Our FSMs include interceptors for frequently encountered dynamic behaviors such as cookie consent dialogs and permission prompts (see complexity_profile in Appendix A.1). These are among the most common "unexpected" interactions agents face on real websites.
>
> **(3) Dynamic data updates can be integrated.** The reviewer's suggestion about asynchronous updates is valuable. Within our framework, we can simulate real-time data changes by periodically regenerating the underlying data.js, so the agent encounters different content across episodes while the FSM transition logic remains intact. We plan to explore this extension.
>
> **(4) Empirical transfer is strong.** Despite FSM limitations, our model achieves state-of-the-art on WebVoyager—a benchmark of real websites with dynamic search, lazy loading, and real-time content. This confirms that FSM-trained agents learn robust interaction patterns that generalize beyond deterministic settings.
>
> ## Key Questions
>
> **Q1: AutoWebWorld vs Explorer?** See W1 above for the direct comparison.
>
> **Q2: Transfer to dynamic real-world websites?** We categorize the 9 WebVoyager websites by their primary dynamic features:
>
> | Dynamic Feature | Websites | Ours (%) | Qwen2.5-7B (%) |
> |---|---|---|---|
> | Dynamic search & autocomplete | Apple, Arxiv, Coursera, HF | 23.22 | 5.13 |
> | Conditional form rendering | GF, GM | 3.75 | 0.0 |
> | Real-time / updated content | BBC, WA | 30.71 | 5.30 |
> | Structured lookup | CD | 72.09 | 16.28 |
>
> Our model shows strong improvements across all categories including highly dynamic ones, with no systematic degradation on websites involving heavier dynamic behavior.
>
> **Q3: How realistic are the FSM environments?** We assess realism from two angles:
>
> *Task difficulty.* Table 6 shows frontier agents achieve *lower* success rates on AutoWebWorld (UI-TARS: 20%, Claude-4-Sonnet: 16%) than on WebVoyager (26.51%/26.11%), confirming non-trivial interaction challenges.
>
> *Visual realism.* We captured homepage screenshots from 13 synthesized websites and their real-world counterparts, then measured similarity using two metrics: CLIP image embedding cosine similarity and Gemini-3-Flash pairwise visual scoring (0–1 scale).
>
> | Site | CLIP | VLM |
> |---|---|---|
> | amazon | 0.6544 | 0.885 |
> | coursera | 0.6315 | 0.85 |
> | discord | 0.7967 | 0.62 |
> | github | 0.9092 | 0.735 |
> | medium | 0.6276 | 0.88 |
> | outlook | 0.8836 | 0.75 |
> | quora | 0.7021 | 0.88 |
> | slack | 0.5876 | 0.78 |
> | spotify | 0.6929 | 0.82 |
> | stackoverflow | 0.7412 | 0.75 |
> | walmart | 0.7838 | 0.89 |
> | youtube | 0.7436 | 0.88 |
> | **Average** | **0.7295** | **0.81** |
>
> Notably, our current pipeline only provides the reference website name as text to the coding agent—no reference screenshots are used as input. Newest experiments show that with reference homepage images further improves visual similarity, indicating substantial room for improvement.

---

> > ### Author Rebuttal · Reviewer_5Uut · 2026-04-03
> >
> > Thank you for your rebuttal. The new experiments addressed my main concern regarding the paper’s central motivation, and I will raise my score accordingly.

---

> > > ### Author Response · Authors · 2026-04-05
> > >
> > > We sincerely thank you for the positive acknowledgment and for raising your score. Your detailed feedback during the review process was invaluable in strengthening our work

---

### Decision · Program_Chairs · 2026-04-30

**Decision:**

Accept (regular)

**Comment:**

This paper proposes AutoWebWorld, a framework for generating scalable and verifiable web-agent training data by modeling web environments as finite state machines (FSMs). The system synthesizes interactive websites from FSM specifications and automatically generates verified trajectories. Training web agents on these synthetic trajectories improves performance on several real-world benchmarks, including WebVoyager.

Strengths: Reviewers agree that the paper addresses an important bottleneck in web-agent training, namely the difficulty of collecting large-scale trajectories from real web websites. The proposed formulation of web environments as FSMs provides a clean and conceptually appealing mechanism for intrinsic verification, eliminating reliance on external judges during data generation. The end-to-end automated pipeline and the empirical improvements on WebVoyager and related benchmarks demonstrate the practical usefulness of the approach.

Weaknesses and Remaining Concerns: Some reviewers raised concerns about the synthetic-to-real gap: FSM-based environments may fail to capture complex behaviors of real websites, including dynamic content, asynchronous updates, and other sources of unpredictability. Questions were also raised regarding evaluation methodology, including the use of LLM-based judges for WebVoyager and the limited subset of websites used for evaluation. In addition, one reviewer noted missing statistical reporting (e.g., multi-seed experiments). I agree with these concerns, and urge the paper to conduct human evaluation on the Online-Mind2Web benchmark or using more advanced LLM-based judges (such as WebJudge as in Online-Mind2Web rather than a plain LLM like Gemini or report the judge's alignment with human evaluation). The rebuttal addressed several of these concerns by adding additional experiments and clarifications, and reviewers indicated that their main concerns were resolved.

Overall, reviewers agreed that the paper presents a novel and useful approach to generating verifiable training data for web agents, with promising empirical results and a new research direction. While limitations remain regarding the abstraction gap between synthetic and real environments, the rebuttal sufficiently clarified the method and strengthened the experimental evidence.